# Anomalous electrons in a metallic kagome ferromagnet

Sandy Adhitia Ekahana[1], Y. Soh[1 ✉], Anna Tamai[2], Daniel Gosálbez-Martínez[3,4,5,6], Mengyu Yao[1,7], Andrew Hunter[2], Wenhui Fan[8,9], Yihao Wang[10], Junbo Li[10], Armin Kleibert[1], C. A. F. Vaz[1], Junzhang Ma[1,11], Hyungjun Lee[3], Yimin Xiong[10,12,13], Oleg V. Yazyev[3,4], Felix Baumberger[1,2], Ming Shi[1,16] & G. Aeppli[1,3,14,15]

Ordinary metals contain electron liquids within well-defined 'Fermi' surfaces at which the electrons behave as if they were non-interacting. In the absence of transitions to entirely new phases such as insulators or superconductors, interactions between electrons induce scattering that is quadratic in the deviation of the binding energy from the Fermi level. A long-standing puzzle is that certain materials do not fit this 'Fermi liquid' description. A common feature is strong interactions between electrons relative to their kinetic energies. One route to this regime is special lattices to reduce the electron kinetic energies. Twisted bilayer graphene[1-4] is an example, and trihexagonal tiling lattices (triangular 'kagome'), with all corner sites removed on a 2 × 2 superlattice, can also host narrow electron bands[5] for which interaction effects would be enhanced. Here we describe spectroscopy revealing non-Fermi-liquid behaviour for the ferromagnetic kagome metal $Fe_3Sn_2$ (ref. 6). We discover three $C_3$-symmetric electron pockets at the Brillouin zone centre, two of which are expected from density functional theory. The third and most sharply defined band emerges at low temperatures and binding energies by means of fractionalization of one of the other two, most likely on the account of enhanced electron–electron interactions owing to a flat band predicted to lie just above the Fermi level. Our discovery opens the topic of how such many-body physics involving flat bands[7,8] could differ depending on whether they arise from lattice geometry or from strongly localized atomic orbitals[9,10].

For interactions between electrons to matter in the sense of yielding departures from the conventional theory of metals, the interactions must be strong relative to the electron kinetic energy or bandwidth ($W$). Therefore, it is natural to consider systems characterized by intrinsically small $W$, corresponding to nearly flat bands for the electrons. There are two material classes meeting this requirement, the first containing atoms with electrons that can occupy orbitals with small overlap, resulting in small hopping probabilities and narrow bands. Materials of the second class are built from lattices for which destructive interference between different hopping paths reduces $W$, even though the underlying hopping amplitudes are large. The latter approach was proposed for the kagome lattice, alongside the prediction that, if the host material were also a ferromagnetic metal, it could show a fractional quantum Hall effect in the absence of an external magnetic field[11]. Rhombohedral (space group $R\bar{3}m$, SG-166) $Fe_3Sn_2$ is precisely such a substance[12]. Its building blocks are bilayers of distorted,

three-fold (not six-fold) symmetric kagome layers of Fe atoms alternating with single layers of stanene, as shown in Fig. 1a. It is also a ferromagnet with a high Curie temperature (640 K), which undergoes a first-order spin-reorientation transition near 120 K (refs. 13,14).

The $Fe_3Sn_2$ electronic states have been investigated by various spectroscopies[15-19] and bulk-transport measurements[13,20-23]. The simplest tight-binding calculations (single orbital per site of kagome lattice) for idealized 2D slabs reveal flat bands and Dirac points. By contrast, density functional theory (DFT) for the bulk material yields neither obvious flat bands nor Dirac nodes (Extended Data Figs. 1–3) but instead Weyl points near the Fermi surface (FS)[15]. DFT further accounts for the seeming appearance of Dirac nodes in angle-resolved photoemission spectroscopy (ARPES)[15,18] by suggesting that they[15,16] derive from a coincidence between minima and maxima of surface and bulk bands, respectively. The ARPES data collected so far show neither resolved flat bands nor sharply defined Weyl nodes. However, there is another

[1]Paul Scherrer Institute, Villigen, Switzerland. [2]Department of Quantum Matter Physics, University of Geneva, Geneva, Switzerland. [3]Institut de Physique, École Polytechnique Fédérale de Lausanne (EPFL), Lausanne, Switzerland. [4]National Centre for Computational Design and Discovery of Novel Materials (MARVEL), École Polytechnique Fédérale de Lausanne (EPFL), Lausanne, Switzerland. [5]Departamento de Física Aplicada, Universidad de Alicante, Alicante, Spain. [6]Instituto Universitario de Materiales de Alicante (IUMA), Universidad de Alicante, Alicante, Spain. [7]Max Planck Institute for Chemical Physics of Solids, Dresden, Germany. [8]Beijing National Laboratory for Condensed Matter Physics, Institute of Physics, Chinese Academy of Sciences, Beijing, China. [9]University of Chinese Academy of Sciences, Beijing, China. [10]Anhui Province Key Laboratory of Condensed Matter Physics at Extreme Conditions, High Magnetic Field Laboratory of the Chinese Academy of Sciences, Hefei, China. [11]Department of Physics, City University of Hong Kong, Kowloon Tong, Hong Kong SAR. [12]Department of Physics, School of Physics and Optoelectronics Engineering, Anhui University, Hefei, China. [13]Hefei National Laboratory, Hefei, China. [14]Department of Physics, Eidgenössische Technische Hochschule Zürich (ETH Zürich), Zurich, Switzerland. [15]Quantum Center, Eidgenössische Technische Hochschule Zürich (ETH Zürich), Zurich, Switzerland. [16]Present address: Center for Correlated Matter and School of Physics, Zhejiang University, Hangzhou, China. ✉e-mail: yona.soh@psi.ch

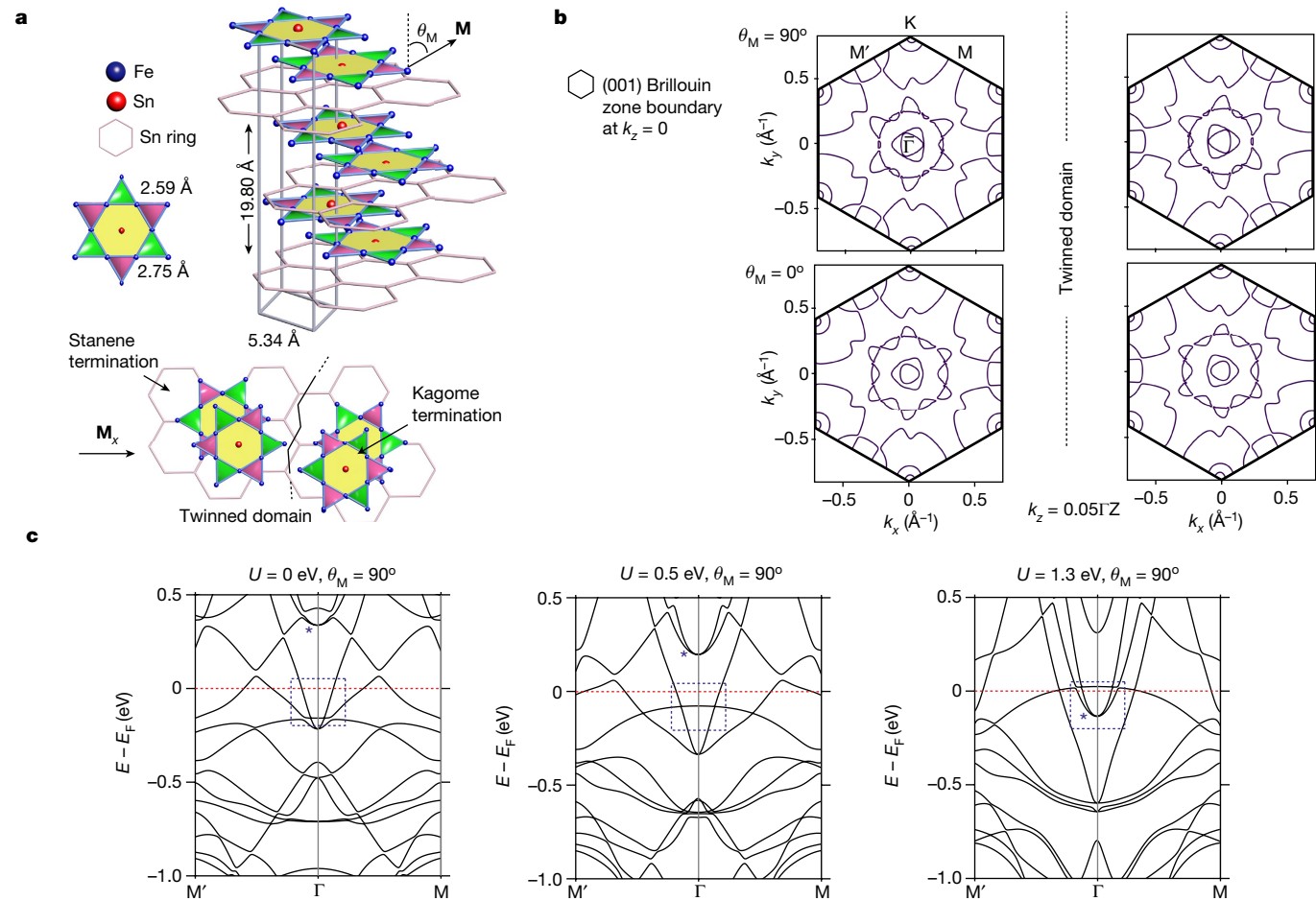

**Fig. 1 | Crystal structure and calculated band structure of Fe₃Sn₂. a**, Crystal structure of Fe₃Sn₂ showing a stanene layer sandwiched between kagome bilayers. $\theta_M$ denotes the angle between the magnetic moment **M** and the *c* axis. Twin domains may occur when the system is rotated around the *c* axis. **b**, Calculated FS at $k_z = 0.05\Gamma Z$ for $U = 1.3$ eV showing a three-fold pattern (left)

and its twinned image (right) when the magnetic moment is pointing transverse to the *c* axis ($\theta_M = 90°$; top) and along the *c* axis ($\theta_M = 0°$; bottom). **c**, Results of DFT calculations for $k_z = 0$ and magnetization in the plane with representative $U = 0$ eV, 0.5 eV and 1.3 eV revealing an electron pocket (denoted by an asterisk), which is investigated by μ-ARPES (marked by the area within the dashed box).

feature that unambiguously distinguishes the tight-binding calculations from DFT predictions, namely, that only DFT predicts electron pockets surrounding the zone centres[15] (Fig. 1c and Extended Data Figs. 1–3). Furthermore, as shown in Fig. 1c, the pockets are highly sensitive to electron-correlation effects as parametrized by the effective Coulomb interaction *U* in DFT+*U* calculations, and so could be exploited for both measurement of *U* as well as the discovery of new electronic phenomena in which the effective band structure is strongly influenced by the Coulomb interactions encoded by *U*.

Here we report the first, to our knowledge, detailed experimental study of the distinct electron pockets, by using state-of-the-art laser-based micro-focused ARPES (μ-ARPES) to overcome averaging over crystallographic twins as well as the surface sensitivity of conventional synchrotron-based ARPES.

## Motivation for spatially resolved ARPES experiments

ARPES is the natural technique for determining the electronic band structure and has been applied on numerous occasions to Fe₃Sn₂ (refs. 15–18). For this material, ARPES faces potential difficulties beyond the usual challenge of distinguishing surfaces and bulk contributions. The first relates to the large number of atoms per unit cell (18 Fe and 12 Sn), leading to closely spaced bands, which—owing to magnetic order—will also be spin split. The second difficulty is the possibility of crystal twinning associated with the 'breathing' kagome motif of Fe₃Sn₂

(Fig. 1a), which corresponds to rotations by $\delta = m \times 60°$, $m = \pm(1,3,5)$ (Fig. 1a,b). The third is the presence of ferromagnetism, in which—owing to spin–orbit coupling (SOC)—different magnetic domains are associated with different electronic structures[15]. Ferromagnetism with non-zero in-plane-ordered moment breaks rotation symmetry, although difficult to discern compared to the overall three-fold symmetry of the underlying crystal structure[15]. However, the ARPES images published so far are predominantly six-fold symmetric[15–18], which can only be attained on the high-symmetry $k_z = \left(0, \frac{3\pi}{c}\right)$ planes for a single crystallographic domain with no in-plane ferromagnetism. Tanaka et al.[18] interpreted their ARPES results as three-fold symmetric for $k_z \neq \left(0, \frac{3\pi}{c}\right)$, but their data were not sufficient to reach that conclusion unambiguously. These difficulties can only be remedied by high-energy-resolution μ-ARPES, with spatial resolution sufficient to resolve magnetic and structural domains over which ordinary ARPES measurements perform averages. As described in Methods, we have also used a variety of X-ray-based microscopies to ensure that the laser μ-ARPES instrument with its 3 μm (full width at half maximum (FWHM)) spot size meets the spatial resolution requirement.

## Laser μ-ARPES

The 6.01 eV photon energy limits the momentum space probed to near the Γ point, which—nonetheless—is of great interest because it enables detailed imaging of the electron pocket seen previously

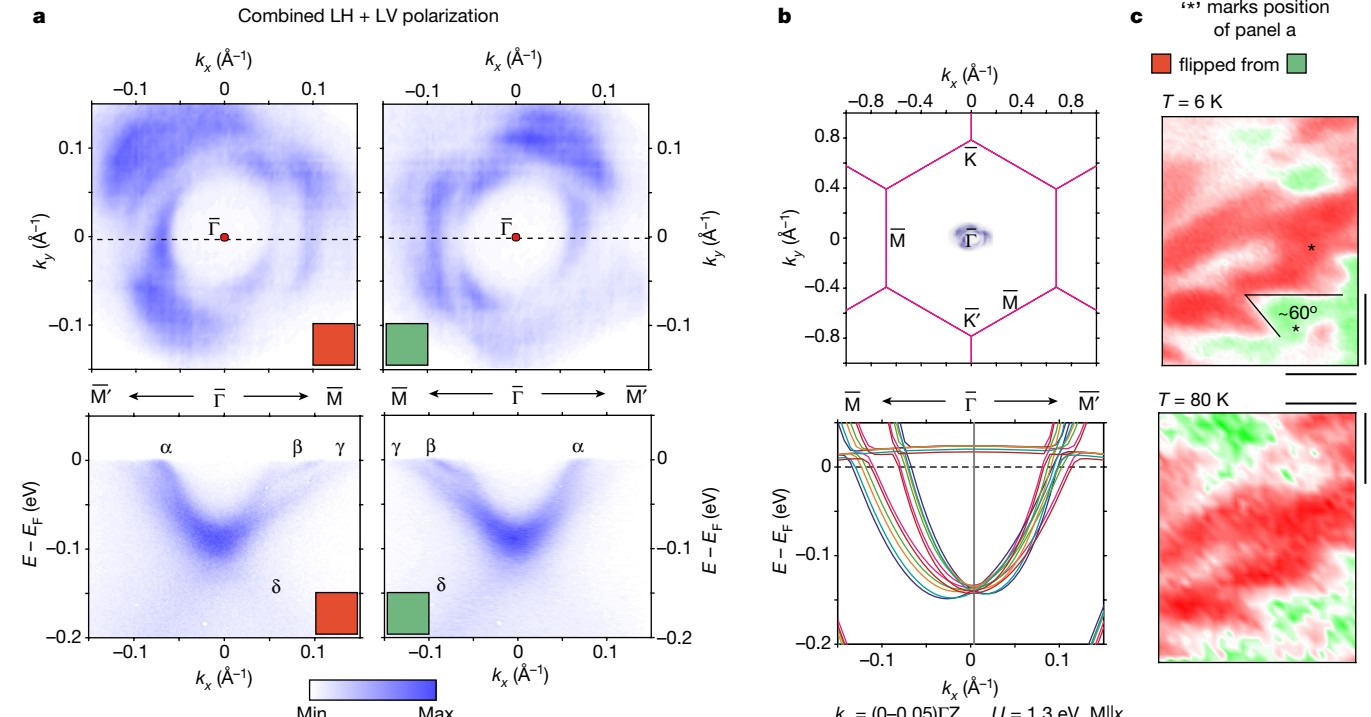

**Fig. 2 | Spatially resolved laser ARPES data and twin domains of Fe₃Sn₂.**
**a**, Laser ARPES data for $Fe_3Sn_2$. Top row, FSs showing a three-fold pattern with an electron pocket at the $\bar{\Gamma}$ point and three petals, which are rotated relative to each other, distinguishing the crystal twin domains. Bottom row, energy versus momentum cut of one of the petals (dashed lines in the plots in the top row, $\overline{M}\bar{\Gamma}\overline{M}'$ direction) showing the electron-pocket band labelled α, a band labelled β, which abruptly disappears through α, a petal labelled γ sharing a bottom

with the α band and, last, a broad and weak band, referred to as δ. **b**, Top, illustration of the reciprocal space probed with laser ARPES, which spans a small portion of the total Brillouin zone. Bottom, DFT calculation along the $\overline{M}\bar{\Gamma}\overline{M}'$ direction for $k_z = (0–0.05)\Gamma Z$ as an illustration of $k_z$ broadening and asymmetrization, agreeing tentatively with the formation of α and γ bands. **c**, Twin domain patterns at 6 K and 80 K. Scale bars, 40 μm.

at low resolution with a large beam[15] and predicted by DFT[15,17,18], but not by published tight-binding models[16]. Figure 2a shows data at 6 K, which reveal this electron pocket with unprecedented clarity for two laser-beam positions on the sample.

The FSs are located on three rings in ($k_x$, $k_y$), with the inner ring clearly three-fold modulated and the second and third rings less obviously so, but nonetheless with their most intense feature 60° out of phase with the inner ring (Fig. 2a). The rings are located near the Brillouin zone centre as in Fig. 2b, top. Comparison of the data in Fig. 2a, top row immediately indicates a relative rotation of 180°, equivalent to 60° for this rhombohedral material. We thus come to the important finding that the electronic structure of $Fe_3Sn_2$ at 6 K shows two distinct yet equivalent areas rotated from each other by 180°. The dispersion curves in Fig. 2a, bottom row show the intercepts of three bands with the three FSs and confirm the same 180° rotation in ($k_x$, $k_y$), which relates to their respective FSs measured in Fig. 2a, top row. Two of the bands define the innermost and outermost FSs and are labelled as α and γ, respectively, in Fig. 2a, bottom row and merge to form a parabola centred at $\bar{\Gamma}$ with pocket minimum at about 0.085 eV. There is also a very sharp feature between α and γ, labelled β, which abruptly loses intensity around $E_B = 0.025$ eV, as can be seen from the energy–momentum cut image (Fig. 2a, bottom row). We also notice a very weak broad band at higher binding energy, labelled as δ.

The two distinct areas rotated by 180° with respect to each other, that is, twinned domains, can be mapped in real space by colour coding the intensity asymmetry between the left and right sides of the α band in the ARPES cut (Fig. 2a, bottom row). In the upper panel of Fig. 2c, we see that there is a relatively abrupt transition from the 'red' to 'green' regions, indicating that the twinned domain is rotated discontinuously.

A rough measurement of the domain-wall angle gives a value of 60°, in agreement with the twin rotation for the crystal structure.

The bulk electronic properties of $Fe_3Sn_2$ undergo pronounced evolution on cooling below 80 K, at which the spin reorientation is complete[13]. Particularly notable are the modulation of the carrier density depending on the magnetization direction below 80 K (ref. 23) and the temperature dependence of the anisotropic magnetoresistance and planar Hall effect[22], which features a three-fold (not six-fold) antisymmetric term as a function of in-plane field direction. To determine whether any of the ARPES features that we see at low temperature could be related to these transport phenomena, we have collected μ-ARPES data at 80 K, which is near the onset of the unusual magnetoelectrical phenomena but still in the regime in which the magnetism is dominated by domains with moments along the kagome planes[14]. The bottom panel of Fig. 2c shows the intensity asymmetry image for 80 K, which is not appreciably different from the 6 K image above it. The main effect of warming is a broadening of the spectra and, in particular, the loss of the very sharp β band, as seen in Fig. 3a,ii,iv. We attribute the three-fold pattern itself to probing (slightly) off the high-symmetry $k_z = 0$ plane (see Methods for further explanation), consistent with our bulk DFT calculations; high-symmetry probing at $k_z = 0$ or $k_z = \frac{3\pi}{c}$ will generate a six-fold pattern regardless of the 'breathing' of the kagome layers.

## Comparison with DFT+*U* calculations

In the calculations, electron pockets centred on Γ and close to the Fermi level, $E_F$, exist for $U = 0–3.0$ eV (Extended Data Fig. 1). However, the distance $\Delta E$ to $E_F$ from the bottom of the bands for the pocket varies strongly with $U$, as do the Fermi vectors $k_F$, which can be seen in Fig. 1c

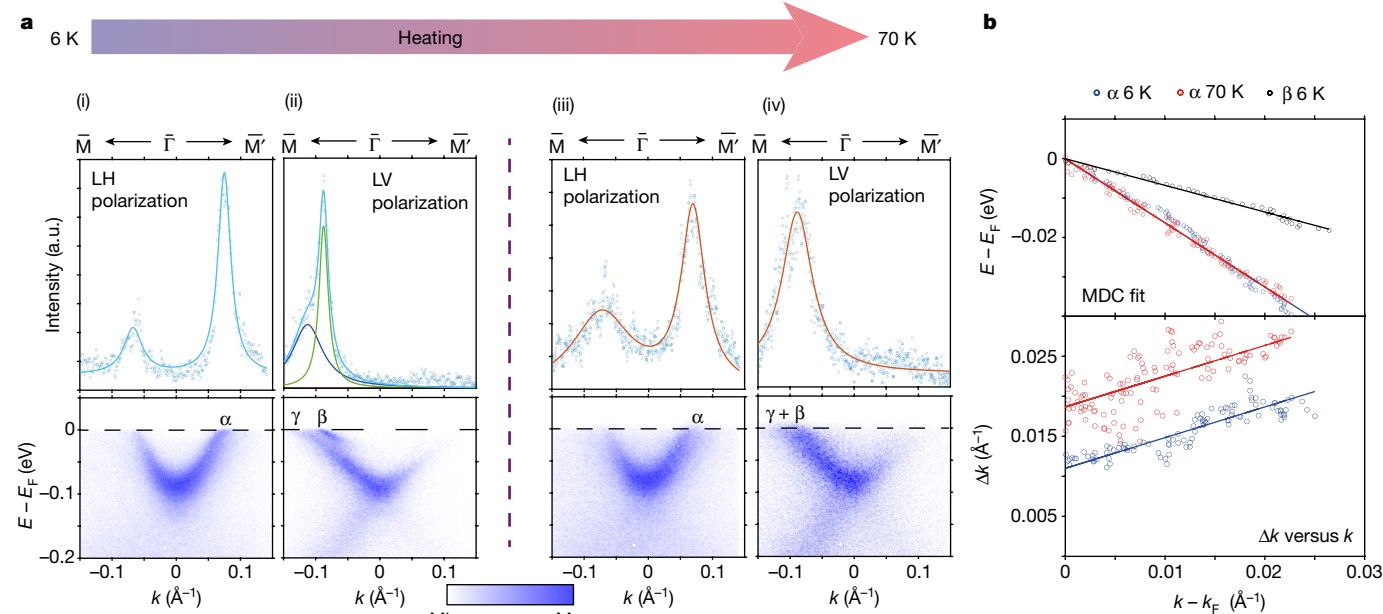

**Fig. 3 | Fitting of MDCs and extraction of peak positions and widths. a**, The $\overline{\text{M}}\,\overline{\Gamma}\,\overline{\text{M}}'$ cut from LH polarization shows mostly the band labelled α at 6 K (i) and 70 K (iii), with the MDC at $E_F$ and the fitting result. The $\overline{\text{M}}\,\overline{\Gamma}\,\overline{\text{M}}'$ cut from LV polarization shows mostly the bands labelled β and γ at 6 K (ii) and 70 K (iv), with the MDC at $E_F$ and the fitting result. **b**, Top, plot of the fitted peak position from the MDC for α (blue 6 K and red 70 K) and β (black 6 K) with resulting linear fit superposed. Bottom, plot of peak width Δk versus $k - k_F$, showing linear behaviour with same gradient close to $E_F$ for α (blue 6 K and red 70 K) consistent with the marginal Fermi liquid hypothesis. On heating to 70 K, the line is shifted as described by equation (2).

and Extended Data Figs. 1–3. The best description of the α and γ bands, taking into account both the band bottom and $k_F$, occurs for $U = 1.3$ eV, differing from the values $U = 0.5$ eV (ref. 15) or $U > 2$ eV (ref. 24) used previously and more than ten times larger than the measured band-width $W \approx 0.1$ eV for the electron pocket.

We show the DFT cut along $\overline{\text{M}}\,\overline{\Gamma}\,\overline{\text{M}}'$ by combining the $k_z = (0–0.05)$ ΓZ, $U = 1.3$ eV in Fig. 2b, bottom. The cut is dominated by parabolic bands coexisting with a relatively flat band located around 15 meV above $E_F$ and supports the assertion that α and γ originate from bulk states smeared because of $k_z$ broadening[25] and are asymmetric (three-fold instead of six-fold) owing to slight off $k_z = 0$ probing. We have labelled the features in the cut accordingly: the $k_z$ broadening yields two visible branches cutting $E_F$ along the $\overline{\text{M}}\,\overline{\Gamma}$ trajectory, whereas there is a broader, merged α-like dispersion along $\overline{\Gamma}\,\overline{\text{M}}'$. The Fermi velocities do not precisely match those obtained by ARPES but can be tuned through small adjustments to $E_F$ on account of the apparent hybridization with the flat band just above $E_F$. We note that, irrespective of the choice of $U$, substantial differences remain between ARPES and DFT. None of the existing DFT calculations reproduces the very small gap between the bottom of the α and γ bands and the top of δ; furthermore, the sharp β band found by ARPES at low T near $E_F$ is not seen in calculations.

## Sharp quasiparticles at low temperatures

Experimental data shown in Fig. 2a yield key parameters, including FS areas, velocities and scattering lengths, as summarized in Methods. The β-band FS centred on $\overline{\Gamma}$ has area equivalent to a de Haas–van Alphen (dHvA) frequency $n_{\text{dHvA}} = 254 \pm 20$ T, in reasonable agreement with that reported in ref. 26 (approximately 200 T) and attributed to the 'Dirac ring' emanating from the putative Dirac point at $\overline{\text{K}}$ and 0.1 eV below $E_F$. Given that DFT accounts for electron pockets centred on Γ while explaining the Dirac-like features in terms of an accidental overlap of surface and bulk bands[15,18], a reinterpretation of the observed $n_{\text{dHvA}}$ is needed.

For the β band, the measured quasiparticle scattering length $\lambda = \frac{1}{\Delta k} = 155 \pm 5$ Å, in which Δk is the half width of the momentum distribution curve (MDC) peaks (at 6 K and at $E_F$), is remarkably long for a topological (bulk) metal and comparable with that for the much simpler surface Dirac quasiparticles for the n-type $Bi_2Te_3$ topological insulator (about 190 Å)[27]. For α, λ is reduced by a factor of two to around 85 Å but is still long by the same standards. These values are consistent with the residual ($T = 2$ K) resistivity $\rho_{xx}(0) = 4.35$ μΩ cm and a carrier density of $1.8 \times 10^{28}$ m$^{-3}$ from transport data[13,23] for similar samples. Applying the Drude formula $\sigma_0 = \frac{ne^2\tau}{m}$ for conductivity, we obtain a scattering time $\tau \approx 4.5 \times 10^{-14}$ s by assuming that the carriers have an effective mass $m = m_e$. For an averaged Fermi velocity of about 1.2 eVÅ, τ converts to a mean free path $l = v_F\tau \approx 82$ Å, which is on the order of the ARPES results for the α and β bands.

## β-band formation

The most notable feature of the β band is its appearance at the very modest binding energy of roughly 25 meV as well as on cooling, both evident in Fig. 3a as well as in Fig. 4a,b, showing higher statistics data for another sample. The qualitative outcome was usually the same for all crystals, but with different FS parameters: the essential feature of the β band is that, although its $k_F$ is smaller, its Fermi velocity is the same as that of the γ band. This suggests that the two bands are linked, a hypothesis that we tested further by the spectral analysis shown in Fig. 4c,d. The MDC peak areas in Fig. 4d are normalized by the integrated area of the γ-band pocket minimum (the γ-band pocket minimum is not affected by the temperature) and we see an abrupt appearance of the β band as the binding energy falls below 25 meV. What the analysis also shows is that, when the β band disappears, it is accompanied by an increase in the γ-band spectral weight at the same binding energies. On warming to 25 K, the cross-over linked to β-band formation moves to lower binding energy and becomes broader. Notably, the summation of both β-band and γ-band spectral weights is independent of temperature and shows much smoother (indeed nearly flat) behaviour, indicating

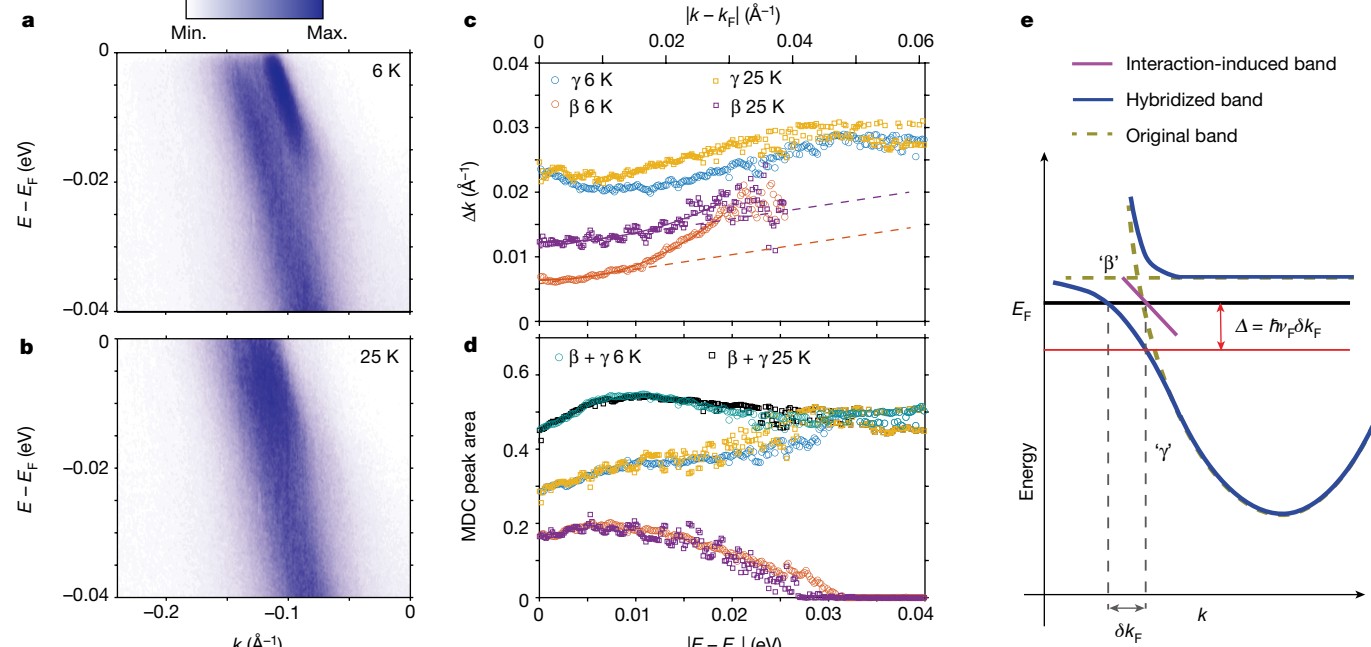

**Fig. 4 | Spectral-weight analysis of β and γ bands and illustration of band hybridization and localization giving rise to the β band. a,b,** High statistics cut along $\overline{M}\,\overline{\Gamma}\,\overline{M}'$ from LV polarization focusing on the β and γ bands at 6 K (**a**) and 25 K (**b**) showing that, at 25 K, the β band broadens but can still be separated from the γ band by fitting the MDC. **c,** The β peak MDC width, Δ*k* versus |*k* − *k*F| from 6 K to 25 K shows a clear shift while keeping the gradient (dashed lines) the same. The solid line represents the phenomenological fit explained by equation (1). **d,** The MDC peak area for β, γ and the sum of both for 6 K and 25 K, showing the spectral-weight distribution between the β and γ bands. As the β-band spectral weight is depleted at binding energies larger

than about 20 meV, spectral weight is gained by the γ band. The spectral-weight function shows that the β band disappears at higher binding energy at 6 K than at 25 K. By contrast, the sum of both spectral weights is independent of temperature and relatively constant versus binding energy, indicating the conservation of the spectral weights. **e,** Illustration of the band hybridization that produces interaction-induced (Mott-like) partial localization, resulting in the appearance of the sharp β band, as described in the text. Also, the flattening of the hybridized band 'γ' near $E_F$ could account for the upturn in Δ*k* observed in **c** for small |*k* − *k*F|.

the conservation of spectral weight between β and γ bands and that, therefore, the β band originates or is 'borrowed' from the γ band.

The β band, unlike the much broader γ band, does not continue to a minimum near $\overline{\Gamma}$. It simply looks like a sharp copy of the γ band displaced upwards by about 20 meV towards $E_F$ (the same energy at which the β band starts to disappear). These observations, not easily accounted for by DFT, suggest a many-body origin for the β band. A possible scenario is that there is strong hybridization between flat-band and γ-band states near the bare FS, which results in interaction-induced (Mott-like) localization that reduces the free carrier density; $k_F$ for the electron pockets seen here (γ) would then be correspondingly reduced by $\delta k_F$, and the resulting (β) band will be raised towards $E_F$ by $\Delta = \hbar v_F \delta k_F$ (Fig. 4e). There is still a broad spectral weight from the unrenormalized (γ) bands at the original $k_F$ and, for binding energies exceeding $\Delta$, the localized electrons rejoin the Fermi sea and the spectral weight will reside primarily near the γ band, as might be calculated using DFT. Warming would similarly act to delocalize the carriers and thus cause the loss of the β 'band' in a gradual manner, as shown by the 25 K data in Fig. 4, towards the indistinguishability of β and γ, as shown in the 70 K data in Fig. 3.

## Energy- and temperature-dependent scattering

The spectral-weight-conserving appearance of a sharp resonance as described above is difficult to account for within ordinary band theory and suggests strong interaction effects. A more subtle manifestation of such effects is anomalous quasiparticle scattering as binding energies dip below $E_F$, seen most famously for the copper oxide superconductors. In particular, the ordinary theory of Fermi liquids asserts that the inverse scattering lengths Δ*k*(*E*), proportional to quasiparticle scattering rates by means of the Fermi velocity, are

$$\Delta k(E) = bk_F \left| \frac{(E - E_F)}{\hbar v_F k_F} \right|^\zeta = bk_F \left| \frac{k}{k_F} - 1 \right|^\zeta \qquad (1)$$

with $\zeta = 2$ and *b* a dimensionless constant, whereas a common finding for 'strongly correlated' matter is $\zeta = 1$, which defines the marginal Fermi liquid ansatz[28,29]. Experiments have yielded a variety of values for *b* in strongly correlated systems for which $\zeta = 1$ (refs. 29–32). For the α band, which is sharply defined and does not seem to be linked to another band (as the β band is to the γ band), $\zeta = 1$ provides a description more consistent with our data (Fig. 3b, bottom) than the simple quadratic form with $\zeta = 2$. More important is that $b_\alpha = 0.353 \pm 0.071$ is close to $b_\pi = \frac{1}{\pi}$, for which the relative quasiparticle wavelength $2\pi/|k(E) - k_F|$ is precisely twice the quasiparticle scattering length $1/\Delta k(E)$, that is, we are almost exactly at the point at which the scattering provides a confining potential for the quasiparticles. On the other hand, for the β band, $b_\beta = 0.137 \pm 0.008$ is much smaller in the small $|E - E_F|$ regime, implying that—somehow—the formation of the β band effectively reduces the scattering. This conclusion is also warranted by the observation that, as we move into the range in which the spectral weight of the β band starts to be depleted, the scattering becomes much stronger, with $b_\beta$ rising to $2.11 \pm 0.10$ when $\zeta = 2.3$ to provide an overall phenomenological fit for the entire range of the β band (Fig. 4c solid fitting line).

For both the α and β bands, warming has a strong impact on Δ*k*(*E*), which—in the linear regime—acts additively, that is

$$\Delta k(E, T) = \Delta k(E, T = 6\,\text{K}) + \Delta k(E = 0, T) \qquad (2)$$

In the cross-over regime in which the β band becomes less visible, Δ*k*(*E*, *T*) is less dependent on temperature.

## Discussion and conclusions

Our combination of high-resolution spectroscopy with single domain selection provides an unprecedented opportunity to examine the quasiparticles in a strongly correlated kagome metal. At low temperature, after elastic scattering contributions are subtracted, the scattering rates for the α band, predicted by DFT, quasiparticles scale with their momenta, implying marginal rather than ordinary Fermi liquid behaviour and associated strong correlation physics. However, the peculiarities of the sharpest band (β), not accounted for by DFT, indicate that $Fe_3Sn_2$ is hosting more than 'conventional' strong correlations. In particular, the β-band spectral weight comes at the expense of diminishing γ weight, raising the profound possibility that, on cooling, electrons are fractionalized between β and γ bands in the approximate ratio of 1:2 based on the spectral weight in Fig. 4d, corresponding to charge 1/3 quasiparticles, but are not lost to charge-density or spin-density wave formation. The flat band seen in DFT just above $E_F$ may be a relevant feature: hybridization of γ with this band could greatly enhance the dimensionless electron–electron interactions beyond the already large $U/W \approx 10$ obtained from our ARPES/DFT work, thus accounting for the sharp and displaced β band, which appears below 70 K together with numerous anomalous magnetotransport properties[13,22,23].

Our discovery of what looks like fractionalized quasiparticles at low temperature in a topological ferromagnetic metal calls for further spectroscopic and electrical measurements as well as theory to understand how the apparent fractionalization of one species of electron is related to the long-standing predictions[7,8,11] of fractional quasiparticles for two-dimensional lattices with flat bands of single-orbital origin. A further question is whether the bands that are flat on account of small underlying orbitals in Anderson and Kondo lattices could also give rise to fractionalization.

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

# Methods

## DFT calculations

**Method.** Electronic-structure calculations are performed within the DFT framework using the projector augmented-wave method[33] as implemented in the Vienna Ab initio Simulation Package (VASP)[34,35]. The exchange-correlation potential is described using the Perdew–Burke–Ernzerhof[36] functional within the generalized-gradient approximation. Plane waves are used as a basis set with a kinetic energy cutoff of 450 eV. To sample the Brillouin zone, we use a $24 \times 24 \times 24$ $\Gamma$-centred $k$-grid. The SOC is included in our calculations as described in ref. 37. We take into account the $p$ and $d$ semi-core states for the Fe and Sn projector augmented-wave datasets used and the simplified on-site Hubbard $U$ correction[38] is added for the Fe $d$ orbitals.

**Flat band and Dirac/Weyl crossing in $Fe_3Sn_2$.** The ARPES data from $Fe_3Sn_2$ reported so far provide no evidence for the flat band predicted from the tight-binding calculations originally proposed to describe this material. We, along with other authors[18], attribute this to the breathing nature of the kagome layers as well as the hopping of the electrons across different kagome layers, that is, the electrons are not confined to the kagome layer only as assumed for the tight-binding calculations.

Dirac crossings at K points in triangular, honeycomb and kagome lattices are common in simple tight-binding calculations. However, these crossings can be gapped by SOC or small perturbations such as 'breathing' distortions of the kagome planes. Thus, our first approach to understanding this kagome material is to use DFT+$U$ for the 3D material without making assumptions about 2D confinement of the electron movement. We adjust the $U$ value to match the ARPES data.

Experimental detection of the Weyl nodes is difficult for ordinary vacuum ultraviolet ARPES because of their large number[15] combined with ferromagnetic and crystallographic domain structures, resulting in a superposition of different band structures, as well as the presence of surface states. Theoretical guidance is also imperfect given that correlation effects can move the nodes.

**'Dirac points', electron pockets and magnetization direction dependence in DFT calculations at various $U$ values.** We attempted to reproduce the 'Dirac points' predicted in the tight-binding model by changing the $U$ value in the DFT calculations as shown in Extended Data Fig. 1 (also Brillouin zone labels). Our results show that the 'Dirac points' are not reproduced for $U$ values between 0 and 1.3 eV. We can see that the closest resemblance of the band dispersion to the linear crossing is achieved at a $U$ value of 0.5 eV or above, at which the gap is narrow but never fully closes.

We determined a Hubbard $U$ value of 1.3 eV to reproduce the ARPES band structure around the $\Gamma$ point at $E_F$. Band structures calculated using several different $U$ values with magnetic moments pointing in the kagome plane are shown in Extended Data Fig. 1c. In Extended Data Fig. 2, the band-structure evolution as a function of magnetic moment direction (M$\|x$, $\theta = 70°$ and M$\|z$) is also plotted with selected $U$ values (0, 0.5 and 1.3 eV). Extended Data Fig. 3 plots the bands calculated in the ARPES energy window for the range $k_z = (0–0.05)\Gamma Z$ (for more details about this, see the section 'Laser µ-ARPES').

## Sample growth and structural characterization

$Fe_3Sn_2$ crystallizes in a rhombohedral structure with space group $R\bar{3}m$, with crystal axis $a,b = 5.34$ Å, $c = 19.80$ Å and $\gamma = 120°$. $Fe_3Sn_2$ single crystals were grown by a vapour transport method. Stoichiometric iron powder (Alfa Aesar, 99.9%) and tin powder (Alfa Aesar, 99.9%) were placed into an evacuated quartz tube. The tube was then annealed at 800 °C for 7 days before quenching in icy water. The crystal structure of the polycrystalline $Fe_3Sn_2$ precursor is confirmed by the X-ray diffraction data shown in Extended Data Fig. 4a. The obtained polycrystalline $Fe_3Sn_2$ precursor was thoroughly ground and sealed with $I_2$ (about 4 mg cm$^{-3}$) in a quartz tube 1 cm in diameter and 16 cm in length. $Fe_3Sn_2$ single crystals were obtained under a temperature gradient of 650 °C (source) to 720 °C (sink) for two weeks.

The cleaved surface for laser ARPES is confirmed to be (001), the $ab$ plane, from low-energy electron diffraction measurements on a freshly cleaved surface, as shown in Extended Data Fig. 4b. The typical cleaved surface is shown in Extended Data Fig. 4c, indicating a flat surface suitable for ARPES measurements.

## Magnetic domain and surface termination probed by XPEEM

We first consider the magnetic domain configuration and different surface terminations, which typical ARPES averages over, by using X-ray photoemission electron microscopy (XPEEM) to obtain spatially resolved X-ray absorption spectroscopy (XAS) and X-ray photoelectron spectroscopy (XPS) maps of the sample using the SPELEEM III instrument (Elmitec GmbH) at the Surfaces/Interfaces: Microscopy (SIM) beamline of the Swiss Light Source. The aim is to establish whether there are variabilities in the sample on the length scale of the laser µ-ARPES experiments. In our case, the XAS is monitored using photoelectrons with kinetic energy below approximately 1–2 eV (low-energy secondary electron, 'bulk' sensitive to a depth of 3–5 nm from the surface), whereas the XPS photoelectrons have a kinetic energy of 96 eV (surface sensitive). The sample was cleaved in vacuum and investigated at 80 K, the base temperature of the cryostat. Extended Data Fig. 5a shows the magnetic domains of the system, which are obtained from the pixel-wise ratio of XAS collected for left circular (CL) and right circular (CR) polarized photons at the Fe $L_3$ edge. Extended Data Fig. 5b shows the map of the photoelectron yield summed over both polarizations (CL + CR), confirming that the iron content in the 'bulk' of the sample is homogeneous.

The surface termination of the $Fe_3Sn_2$ samples is probed by mapping the Sn $3d_{3/2}$ and Fe $2p_{3/2}$ local photoemitted electron intensities. The photon excitation energy was varied such that the kinetic energy of the detected electrons was fixed at 96.8 eV to achieve a high surface sensitivity (about 2 Å according to the universal curve[39]). The photoemission peaks are determined by recording XPEEM image sequences with varying incident photon energy with X-rays linearly polarized in the plane of the sample. Extended Data Fig. 5c shows the XPS map for the same region as in panels a and b, obtained for the Sn 3d levels and indicating variations in the Sn intensity. Similar results are observed for a different cleaved surface, shown in Extended Data Fig. 5f. Panel d shows the intensity histogram from the red square in panel c: it is bimodal (in this case, well characterized as the sum of two Gaussians), which indicates that there are two possible Sn populations for the surfaces formed after cleaving $Fe_3Sn_2$. The ratio between the mean intensities for the two Gaussians is $I_1/I_2 = 1.8 \pm 0.6$, in which $I_1$ is the brighter intensity and $I_2$ is the darker intensity mean value (the derivation is described below). The bimodal distribution of the Sn intensity, also visible in the sharp step in intensity of the line scan shown in panel d, arises because the material consists of (kagome) Fe bilayers alternating with stanene monolayers. In the topmost two layers, one expects either one (one stanene layer) or two Fe layers (no stanene layer), as shown in panel h. Thus, the clear contrast in Sn distribution (panel c) arises from stanene termination for the brighter areas and kagome termination for the smaller, lower intensity islands. Furthermore, given the much bigger area associated with the larger Sn photoemission intensity, we conclude that most of the sample is stanene terminated, agreeing with the suggestion made in ref. 15. Comparison of panels c and a also demonstrates that the magnetic domain pattern is uncorrelated with the surface termination.

A second XPS map is shown in Extended Data Fig. 5e–g, showing a clear cleaved surface and an area with glue residue and a crack. The XPS intensity distributions for the Sn $3d_{3/2}$ and Fe $2p_{3/2}$ peaks are shown in panels f and g, respectively. Electrons emitted from the Sn $3d_{3/2}$ orbitals

have two distinct intensities in the spatially resolved map in panel f, revealing a large region of higher intensity together with smaller islands of lower intensity. By comparison, the iron emission map shown in panel g is more homogeneous and has a lower signal-to-noise ratio owing to the lower photoemission yield of the Fe $2p$ transition. We assign the bright area in panel f as stanene terminated and the darker area as kagome terminated. However, we cannot determine the exact kagome termination, that is, whether it is one or two Fe layers (panel h i or iii) within the intensity resolution of our measurements. A comparison between these two sets of data shows that the area attributed to kagome termination can vary in size and is not limited to the small areas shown in panel c. The XPS spectrum shown in panel g is obtained by integrating over an area larger than that for the Sn XPS peak shown in panel f.

The X-ray magnetic circular dichroism map, collected at $T = 80$ K, reveals a magnetic domain pattern with a hierarchy of characteristic features ranging in size from 50 to <1 μm, indicating the need for μ-ARPES to avoid averaging of momentum and energy-resolved states associated with different magnetization directions. The length-scale distribution is similar to that for previous magnetic force microscopy (MFM) studies[14], which also showed that, at the lower temperatures of our μ-ARPES experiments described below (about 6 K), the magnetic domain pattern was characterized by larger domains and that, although they are smaller at 80 K, they are still predominantly polarized parallel to the kagome planes (domains with perpendicular polarization are a minority and would produce subtly different band structures; see Fig. 1b and ref. 15). At the same time, the different surface terminations as established by XPS are characterized by length scales in the same range, implying that μ-ARPES could also detect differences in surface states should they matter for the photon energy chosen.

**Intensity ratio of the XPS signal.** The photoemission electron micrographs showing different terminations in $Fe_3Sn_2$ can be numerically analysed as follows:

1. We assume that the photon penetration depth is much deeper than the escape depth of the most energetic photoelectron, that is, an electron at $E_F$. Thus, we ignore the exponential decay component of the photon intensity.
2. Assumption (1) leads to an expression for the photoelectron intensity from a given atom called $a$, on layer called $b$, with distance $d$ from the surface as

$$I_{ab} = C_{ab} n_{ab} \exp\left(-\frac{d}{\lambda}\right)$$

in which $\lambda$ is the escape depth of the electron, $C_{ab}$ is the proportionality constant that contains the photon cross-section of atom $a$ at layer of type $b$ and $n_{ab}$ is the atom $a$ density at layer of type $b$.
3. Generalizing (2), we can express the total intensity from atoms $a$ in all layers of type $b$ as a summation

$$I_{ab \text{ total}} = C_{ab} n_{ab} \sum_{i=1}^{\infty} \exp\left(-\frac{d_i}{\lambda}\right)$$

in which we assume the proportionality constant $C_{ab}$ to be the same for all similar layers at different depths.

With these three assumptions, we can obtain the total XPS intensity from both Sn and Fe. First, we notice that there are three possible terminations and call them double kagome termination ($S_{kk}$), single kagome termination ($S_k$) and stanene termination ($S_s$), as shown in Extended Data Fig. 5h.

**Double kagome termination ($S_{kk}$) case.** The total Sn and Fe intensity from this termination can be expressed as

$$
\begin{aligned}
I_{Sn,S_{kk}} &= \sum_{m=0}^{\infty}\left(\exp\left(-m\frac{2z_{ks}+z_{kk}}{\lambda}\right)\exp\left(-\frac{z_{ks}+z_{kk}}{\lambda}\right)C_{Sn,s}n_{Sn,s}\right) \\
&+ \sum_{m=0}^{\infty}\left(\exp\left(-m\frac{2z_{ks}+z_{kk}}{\lambda}\right)C_{Sn,k}n_{Sn,k}\right) \\
&+ \sum_{m=0}^{\infty}\left(\exp\left(-m\frac{2z_{ks}+z_{kk}}{\lambda}\right)\exp\left(-\frac{z_{kk}}{\lambda}\right)C_{Sn,k}n_{Sn,k}\right) \\
&= \frac{\exp\left(-\frac{z_{ks}+z_{kk}}{\lambda}\right)C_{Sn,s}n_{Sn,s}+C_{Sn,k}n_{Sn,k}+\exp\left(-\frac{z_{kk}}{\lambda}\right)C_{Sn,k}n_{Sn,k}}{1-\exp\left(-\frac{2z_{ks}+z_{kk}}{\lambda}\right)}
\end{aligned}
$$

$$
\begin{aligned}
I_{Fe,S_{kk}} &= \sum_{m=0}^{\infty}\left(\exp\left(-m\frac{2h_{ks}+h_{kk}}{\lambda}\right)C_{Fe,k}n_{Fe,k}\right) \\
&+ \sum_{m=0}^{\infty}\left(\exp\left(-m\frac{2h_{ks}+h_{kk}}{\lambda}\right)\exp\left(-\frac{h_{kk}}{\lambda}\right)C_{Fe,k}n_{Fe,k}\right) \\
&= \frac{C_{Fe,k}n_{Fe,k}+\exp\left(-\frac{h_{kk}}{\lambda}\right)C_{Fe,k}n_{Fe,k}}{1-\exp\left(-\frac{2h_{ks}+h_{kk}}{\lambda}\right)}
\end{aligned}
$$

**Single kagome termination ($S_k$) case.**

$$
\begin{aligned}
I_{Sn,S_k} &= \sum_{m=0}^{\infty}\left(\exp\left(-m\frac{2z_{ks}+z_{kk}}{\lambda}\right)\exp\left(-\frac{z_{ks}}{\lambda}\right)C_{Sn,s}n_{Sn,s}\right) \\
&+ \sum_{m=0}^{\infty}\left(\exp\left(-m\frac{2z_{ks}+z_{kk}}{\lambda}\right)\exp\left(-\frac{2z_{ks}}{\lambda}\right)C_{Sn,k}n_{Sn,k}\right) \\
&+ \sum_{m=0}^{\infty}\left(\exp\left(-m\frac{2z_{ks}+z_{kk}}{\lambda}\right)C_{Sn,k}n_{Sn,k}\right) \\
&= \frac{\exp\left(-\frac{z_{ks}}{\lambda}\right)C_{Sn,s}n_{Sn,s}+\exp\left(-\frac{2z_{ks}}{\lambda}\right)C_{Sn,k}n_{Sn,k}+C_{Sn,k}n_{Sn,k}}{1-\exp\left(-\frac{2z_{ks}+z_{kk}}{\lambda}\right)}
\end{aligned}
$$

$$
\begin{aligned}
I_{Fe,S_k} &= \sum_{m=0}^{\infty}\left(\exp\left(-m\frac{2h_{ks}+h_{kk}}{\lambda}\right)\exp\left(-\frac{2h_{ks}}{\lambda}\right)C_{Fe,k}n_{Fe,k}\right) \\
&+ \sum_{m=0}^{\infty}\left(\exp\left(-m\frac{2h_{ks}+h_{kk}}{\lambda}\right)C_{Fe,k}n_{Fe,k}\right) \\
&= \frac{\exp\left(-\frac{2h_{ks}}{\lambda}\right)C_{Fe,k}n_{Fe,k}+C_{Fe,k}n_{Fe,k}}{1-\exp\left(-\frac{2h_{ks}+h_{kk}}{\lambda}\right)}
\end{aligned}
$$

**Stanene termination ($S_s$) case.**

$$
\begin{aligned}
I_{Sn,S_s} &= \sum_{n=0}^{\infty}\left(\exp\left(-n\frac{2z_{ks}+z_{kk}}{\lambda}\right)C_{Sn,s}n_{Sn,s}\right) \\
&+ \sum_{n=0}^{\infty}\left(\exp\left(-n\frac{2z_{ks}+z_{kk}}{\lambda}\right)\exp\left(-\frac{z_{ks}}{\lambda}\right)C_{Sn,k}n_{Sn,k}\right) \\
&+ \sum_{n=0}^{\infty}\left(\exp\left(-n\frac{2z_{ks}+z_{kk}}{\lambda}\right)\exp\left(-\frac{z_{ks}+z_{kk}}{\lambda}\right)C_{Sn,k}n_{Sn,k}\right) \\
&= \frac{C_{Sn,s}n_{Sn,s}+\exp\left(-\frac{z_{ks}}{\lambda}\right)C_{Sn,k}n_{Sn,k}+\exp\left(-\frac{z_{ks}+z_{kk}}{\lambda}\right)C_{Sn,k}n_{Sn,k}}{1-\exp\left(-\frac{2z_{ks}+z_{kk}}{\lambda}\right)}
\end{aligned}
$$

$$
\begin{aligned}
I_{Fe,S_s} &= \sum_{n=0}^{\infty}\left(\exp\left(-n\frac{2h_{ks}+h_{kk}}{\lambda}\right)\exp\left(-\frac{h_{ks}}{\lambda}\right)C_{Fe,k}n_{Fe,k}\right) \\
&+ \sum_{n=0}^{\infty}\left(\exp\left(-n\frac{2h_{ks}+h_{kk}}{\lambda}\right)\exp\left(-\frac{h_{ks}+h_{kk}}{\lambda}\right)C_{Fe,k}n_{Fe,k}\right) \\
&= \frac{\exp\left(-\frac{h_{ks}}{\lambda}\right)C_{Fe,k}n_{Fe,k}+\exp\left(-\frac{h_{ks}+h_{kk}}{\lambda}\right)C_{Fe,k}n_{Fe,k}}{1-\exp\left(-\frac{2h_{ks}+h_{kk}}{\lambda}\right)}
\end{aligned}
$$

in which $C_{Sn,s}$ is the proportionality constant for Sn atoms in the stanene layer, $n_{Sn,s}$ is the Sn density in the stanene layers, $C_{Sn,k}$ and $C_{Fe,k}$ are the proportionality constants for Sn and Fe atoms in the kagome layer, respectively, $n_{Sn,k}$ and $n_{Fe,k}$ are the Sn and Fe densities at the kagome

layer, respectively, $h_{kk}$ is the distance of Fe layers between two adjacent kagome layers (the kagome bilayer), $h_{ks}$ is the distance of Fe layers in the kagome layer to the nearest stanene layer, $z_{kk}$ is the distance of Sn atom layers between two adjacent kagome layers (the kagome bilayer) and $z_{ks}$ is the distance of Sn atom layers from a kagome layer to the nearest stanene layer.

From the expressions above, we can conclude that $I_{Fe,S_s} < I_{Fe,S_k} < I_{Fe,S_{kk}}$ based on the following relation

$$\frac{\exp\left(-\frac{h_{ks}}{\lambda}\right)C_{Fe,k}n_{Fe,k} + \exp\left(-\frac{h_{ks}+h_{kk}}{\lambda}\right)C_{Fe,k}n_{Fe,k}}{1-\exp\left(-\frac{2h_{ks}+h_{kk}}{\lambda}\right)}$$

$$< \frac{\exp\left(-\frac{2h_{ks}}{\lambda}\right)C_{Fe,k}n_{Fe,k} + C_{Fe,k}n_{Fe,k}}{1-\exp\left(-\frac{2h_{ks}+h_{kk}}{\lambda}\right)}$$

$$< \frac{C_{Fe,k}n_{Fe,k} + \exp\left(-\frac{h_{kk}}{\lambda}\right)C_{Fe,k}n_{Fe,k}}{1-\exp\left(-\frac{2h_{ks}+h_{kk}}{\lambda}\right)}$$

According to this model, we should be able to see the Fe peak contrast between different terminations, which we do not identify in our data owing to insufficient signal-to-noise ratio.

Meanwhile, for Sn intensity, we can posit that

$$n_{Sn,s} \approx 2n_{Sn,k}$$

on account of the twice higher density of Sn in the stanene rather than in the kagome layer. It is also reasonable to assume

$$C_{Sn,s} \approx C_{Sn,k}$$

to obtain the double kagome termination ($S_{kk}$) case

$$I_{Sn,S_{kk}} \approx C_{Sn,k}n_{Sn,k}\frac{2\exp\left(-\frac{z_{ks}+z_{kk}}{\lambda}\right)+1+\exp\left(-\frac{z_{kk}}{\lambda}\right)}{1-\exp\left(-\frac{2z_{ks}+z_{kk}}{\lambda}\right)},$$

the single kagome termination ($S_k$) case

$$I_{Sn,S_k} \approx C_{Sn,k}n_{Sn,k}\frac{2\exp\left(-\frac{z_{ks}}{\lambda}\right)+\exp\left(-\frac{2z_{ks}}{\lambda}\right)+1}{1-\exp\left(-\frac{2z_{ks}+z_{kk}}{\lambda}\right)},$$

and the stanene termination ($S_s$) case

$$I_{Sn,S_s} \approx C_{Sn,k}n_{Sn,k}\frac{2+\exp\left(-\frac{z_{ks}}{\lambda}\right)+\exp\left(-\frac{z_{ks}+z_{kk}}{\lambda}\right)}{1-\exp\left(-\frac{2z_{ks}+z_{kk}}{\lambda}\right)}.$$

With these assumptions, we have the Sn intensity relation for different terminations as

$$I_{Sn,S_{kk}} < I_{Sn,S_k} < I_{Sn,S_s}$$

From the intensity relation of both Sn and Fe, we can conclude that they are inversely related to each other.

We investigate the Sn intensity ratio between terminations to correlate with the experimental data we obtained from XPS. In this model, we use

$$z_{ks} \approx 2 \text{ Å and } z_{kk} \approx 2.5 \text{ Å}.$$

The summary of the intensity (proportional to $C_{Sn,k}n_{Sn,k}$) is given in Extended Data Table 1 for two different $\lambda$ values.

Comparing with the experimental result of $I_{bright}/I_{dark} = 1.84 \pm 0.57$ from Extended Data Fig. 5d, although we cannot tell if the darker region is coming from the single or the bilayer kagome, we can safely infer that the bright regions have stanene termination.

## Laser μ-ARPES

For the μ-ARPES measurements, we use a 6.01 eV fourth-harmonic generation continuous laser from LEOS as photon source, a custom-built micro-focusing lens to reduce the beam-spot diameter to about 3 μm and an MB Scientific analyser equipped with a deflection-angle mode to map the dispersion relation while keeping the area of interest intact (that is, not changing owing to sample rotation). Typical energy and angular resolution (FWHM) were 3 meV/0.2°. The pressure during the measurement is kept at <$10^{-10}$ mbar. The sample is mounted on a conventional six-axis ARPES manipulator as described in ref. 40 and the sample position is scanned with an $xyz$ stage of 100 nm resolution and better than 1 μm bidirectional reproducibility. A more detailed description can be found in ref. 41. During the measurement, the temperature is first lowered to 6 K, at which the sample is cleaved in situ, and the sample drift at subsequent higher temperatures is tracked by using the edges of the sample as reference. The samples are pre-aligned to the high-symmetry cut with low-energy electron diffraction after cleaving.

The perpendicular momentum $k_z$ of the electrons measured by ARPES can be obtained from the expression

$$k_z = \sqrt{\frac{2m_e^*}{\hbar^2} \times (K_{out}+V_o) - \frac{2m_e}{\hbar^2}K_{out}\sin^2\phi}$$

in which $m_e^*$ is the effective mass of the electron, $K_{out} = h\nu - w - |E_b|$ is the kinetic energy of the electron, $h\nu$ is the photon energy, $|E_b|$ is the binding energy of the electron, $w$ is the work function of the detector, $V_o$ is the inner potential of the material and $\phi$ is the analyser slit angle (more details can be found in refs. 25,42). The laser photon energy used, 6.01 eV, corresponds to a perpendicular momentum close to the centre of the Brillouin zone $k_z \approx i \times \frac{2\pi}{c}$, in which $i_{6.01eV} \approx 5.93 \approx 6$, as shown in Extended Data Fig. 6a, whose inner potential used is taken from ref. 15 and assuming $m_e^* = m_e$. The synchrotron photon energy used is 48 eV, which also lies roughly in the same position at the centre of the Brillouin zone, $i_{48eV} \approx 11.98 \approx 12$. From Extended Data Fig. 6a, we can see that these two photon energies differ by two Brillouin zones.

We can also estimate the uncertainty ($\delta k_z$) of $k_z$ determination from the escape depth of the electron. In this case, we can focus on the electron close to $E_F$ ($K_{out} \approx 1.64$ eV), which has an escape depth around $\lambda \approx 60$ Å according to the universal curve[39], giving us $\delta k_z = \frac{1}{\lambda} \approx 0.016$ Å$^{-1} \approx$ 0.05ΓZ. In Fig. 2b, bottom, we show DFT calculations for $k_z$ values between 0 and 0.05ΓZ, revealing that, at $E_F$, the dispersion along $k_z$ could easily result in contributions to the in-plane $\Delta k$ of order 0.02 Å$^{-1}$. Furthermore, the γ band has a larger dispersion along ΓZ than the α band, accounting for the larger $\Delta k$ seen in the experiments for the γ band.

**Synchrotron versus laser ARPES.** Crystallographic twins exist in $Fe_3Sn_2$ and both will contribute to the ARPES data if the beam spot is bigger than the size of the domains. For example, we show in Extended Data Fig. 6b an example of ARPES data measured with photon energy 48 eV, temperature 17 K and a spot size of roughly 35 μm collected at the Surface/Interface Spectroscopy (SIS) beamline, Swiss Light Source, equipped with a Scienta R4000 hemispherical analyser. The overall feature shows a six-fold pattern with no indication of a three-fold pattern. A closer look around the centre also shows a rather blurred circular shape, which can be attributed to a combination of all twinned domains that it may cover. This central feature can in fact be reconstructed from the laser ARPES results by combining data from both twinned domains and different polarizations (LH + LV) to give us the picture shown in Extended Data Fig. 6c. Indeed, we reproduce the synchrotron ARPES data but with higher momentum resolution.

**Fitting MDCs.** The MDCs are fitted to obtain the half-peak width at half-maximum ($\Delta k$), which—after correction for the $\delta k_z$ effects described above—is the inverse of the average quasiparticle scattering length. We obtain $\Delta k$ by fitting the MDC peak with a Voigt line shape, that is, a Lorentzian convolved with a Gaussian, for which the Gaussian simulates the instrument response. For momentum scans (MDC), the latter combines angular and energy resolutions to yield a FWHM

$$\delta k \approx \sqrt{(\delta k_x)^2 + (\delta k_y)^2} = \sqrt{(\delta k_x)^2 + \left(\delta E \left(\frac{\partial E}{\partial k}\right)^{-1}\right)^2}$$

in which $v_F = \frac{1}{\hbar}\frac{\partial E}{\partial k}\Big|_F$. In this case, we set $\delta k_x \approx 0.001\ \text{Å}^{-1}$ and $\delta E \approx 4.7$ meV for the data in Fig. 3, $\delta E \approx 1.9$ meV for the data in Fig. 4 and $\frac{\partial E}{\partial k}$ as listed in Extended Data Table 2. We can therefore see that, typically, the broadening in the MDC is owing to the broadening in the energy distribution curve (EDC) because $\delta k_x < \delta E \left(\frac{\partial E}{\partial k}\right)^{-1}$.

**Fitting sharp quasiparticle peak (β) EDC.** Extended Data Fig. 7 shows the sharp quasiparticle peak of β at $E_F$ at 6 K, fitted with a Voigt function in which the Gaussian FWHM was 4.7 meV to represent the detector response (dashed line). We obtain the resolution-corrected Lorentzian FWHM of 2.5 meV.

To fit the 25 K data, we constrain the γ peak position by using the result of the 6 K peak position and let the β peak position at 25 K relax from the maximum separation obtained from the 6 K fitting while keeping the slope of $E(k)$ versus $k$ similar to the γ band.

In Extended Data Table 2, we summarize for the three pockets the Fermi velocity $v_F = \frac{1}{\hbar}\frac{\partial E}{\partial k}$, the area of the electron pocket converted into the dHvA frequency as $f(T) = \frac{\hbar}{2\pi e} \times$ area and the scattering lengths of the quasiparticle bounded from below (on account of $k_z$ broadening[25]) by $\lambda = \frac{1}{\Delta k}$, in which $\Delta k$ is the half width of the MDC peaks (at 6 K and at $E_F$). The area of the electron pockets is calculated by tracing the visible peaks of the corresponding band at $E_F$ and assuming a circular extrapolation to make closed contours.

## Data availability

All data related to this paper are available at a public repository (MARVEL Materials Cloud Archive), with the same title as this paper (https://archive.materialscloud.org).

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

**Acknowledgements** S.A.E., D.G.-M., H.L., O.V.Y. and M.S. acknowledge the support from NCCR MARVEL funded by the Swiss National Science Foundation (SNSF, grant no. 182892). S.A.E. acknowledges the European Union's Horizon 2020 research and innovation programme under the Marie Skłodowska-Curie grant agreement no. 701647. Y.X. acknowledges the National Key Research and Development Program of China (grant no. 2021YFA1600200). S.A.E. and G.A. acknowledge the European Research Council HERO Synergy grant SYG-18 810451. The laser ARPES work at the University of Geneva was supported by the SNSF grants 2000020_165791 and 200020_184998. Part of this work was supported by the High Magnetic Field Laboratory of Anhui Province. Part of this work was performed at the Surfaces/Interfaces: Microscopy (SIM) beamline of the Swiss Light Source, Paul Scherrer Institut, Villigen, Switzerland. All first-principles calculations were performed at the Swiss National Supercomputing Centre (CSCS) under the projects s1146 and mr27. We acknowledge N. Kumar for helping with the X-ray magnetic circular dichroism photoemission electron microscopy measurements, V. Strocov for discussion about $k_z$ broadening in laser ARPES, and M. Müller and R. Moore for helpful discussions.

**Author contributions** G.A. suggested the project. Y.W. and J.L. grew the single crystals, with supervision from Y.X. A.T., M.Y., Y.S., A.H., S.A.E. and G.A. performed laser ARPES, with support from F.B. S.A.E., A.K. and C.A.F.V. performed XPS XPEEM. D.G.-M. performed DFT calculations, with support from H.L. and O.V.Y. S.A.E. and W.F. analysed the ARPES data, with guidance from G.A. and Y.S. and discussions with J.M. and M.S. S.A.E., Y.S. and G.A. wrote the paper. Y.S. supervised the project.

**Funding** Open Access funding provided by Lib4RI – Library for the Research Institutes within the ETH Domain: Eawag, Empa, PSI & WSL.

**Competing interests** The authors declare no competing interests.

**Additional information**
**Correspondence and requests for materials** should be addressed to Y. Soh.

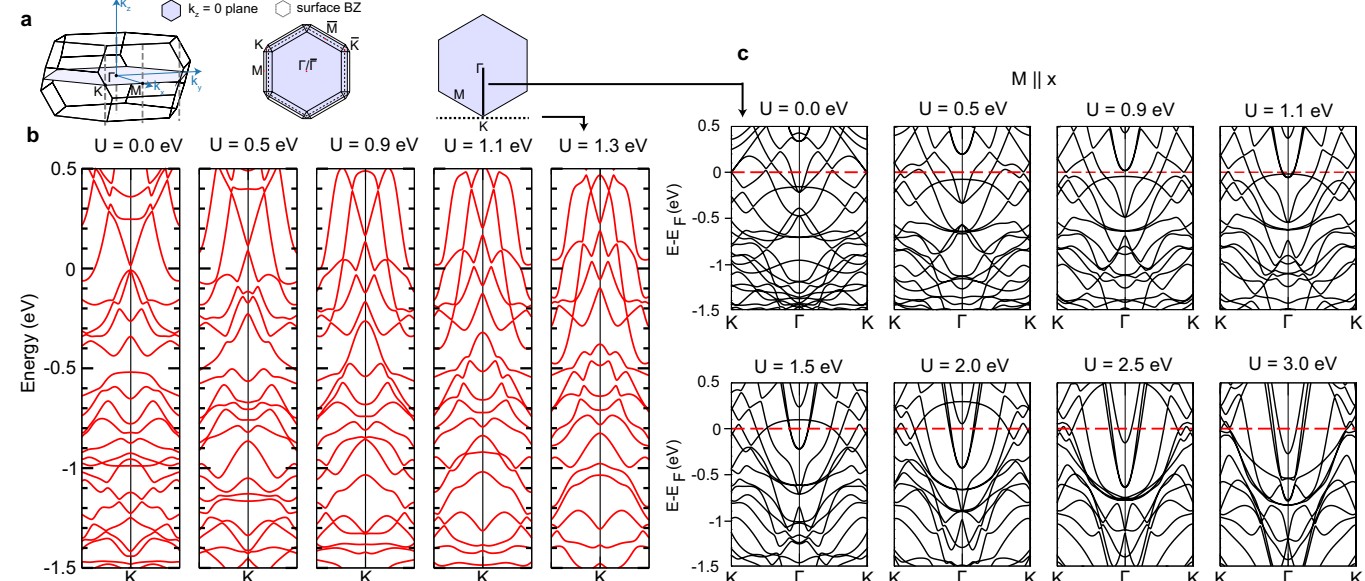

**Extended Data Fig. 1 | Brillouin zone convention and band-structure calculation with varying $U$ values. a**, Brillouin zone of $Fe_3Sn_2$ and the naming convention for this work. **b**, Band-structure calculations of $Fe_3Sn_2$ at $k_z = 0$ performed with different $U$ values fail to reproduce the 'Dirac points' at the

K point along the cut direction shown by the dashed line. **c**, Band-structure plots of $Fe_3Sn_2$ for M∥$x$ and various $U$ values in the direction ΓK at $k_z = 0$, showing the shift of bands, in which the electron pocket close to $E_F$ is formed by different bands.

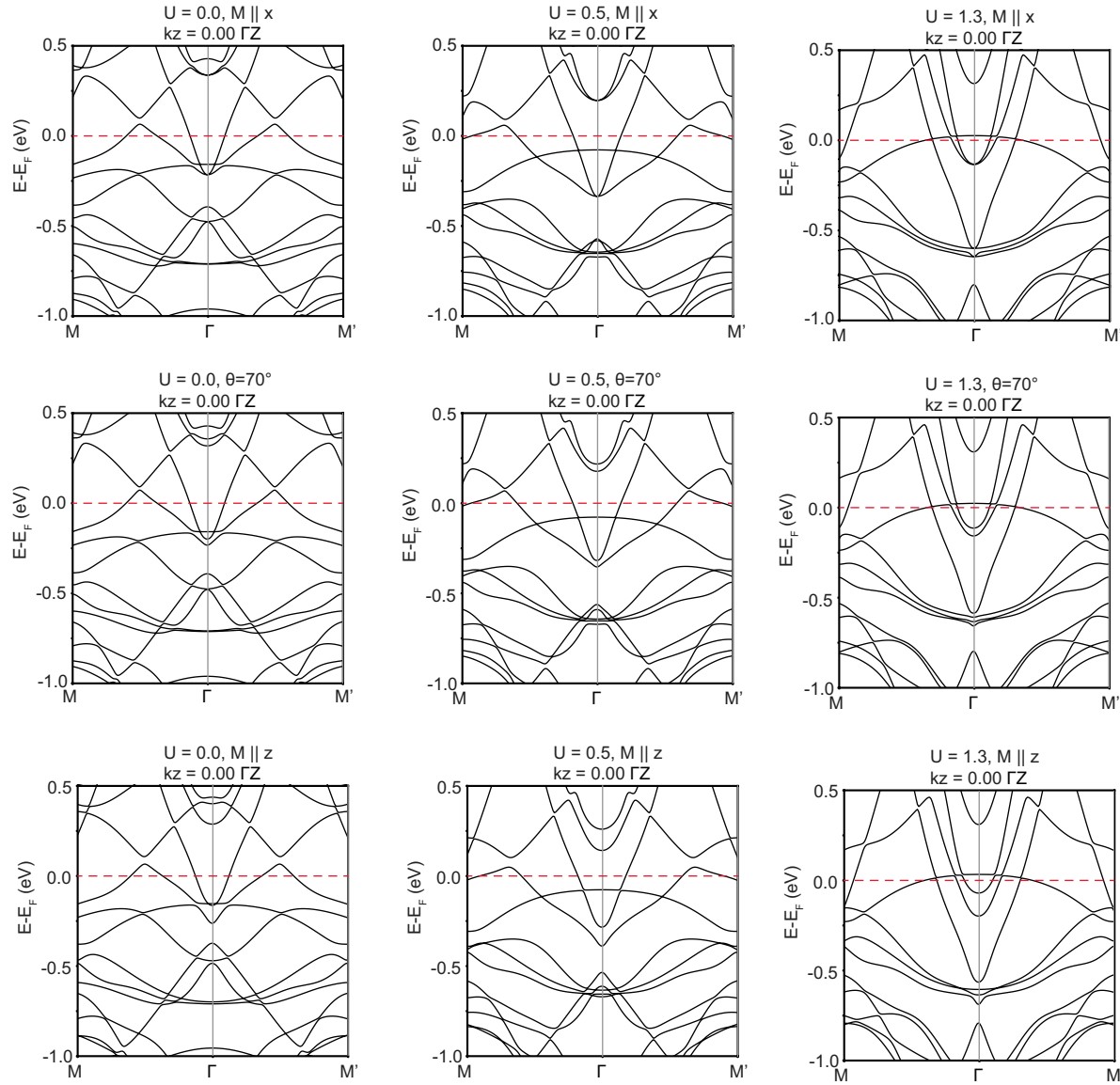

**Extended Data Fig. 2 | Band structure with varying *U* values for various magnetic moment directions.** Band-structure plots of $Fe_3Sn_2$ for selected *U* values (0, 0.5, 1.3 eV) at various moment directions (M∥*x*, $\theta$ = 70° and M∥*z*) in the direction ΓM, showing the shift of bands and band splitting as the moment points more towards the *c* axis.

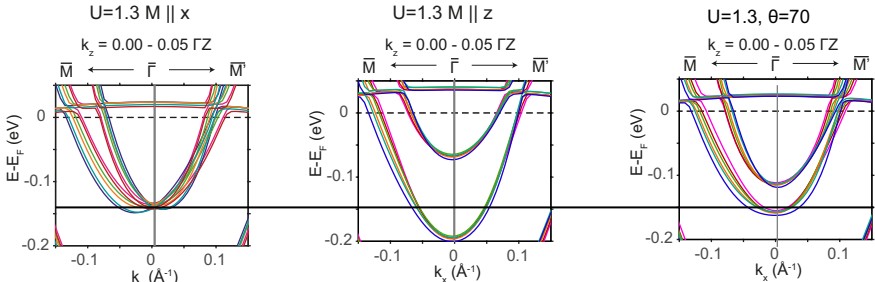

**Extended Data Fig. 3 | Band structure at $U$ = 1.3 eV with varying magnetic moment directions.** Band structure at $U$ = 1.3 eV plot around $k_z$ = (0−0.05)ΓZ (to simulate $k_z$ broadening) for magnetic moment directions along $x$ and $z$ and for $\theta$ = 70°.

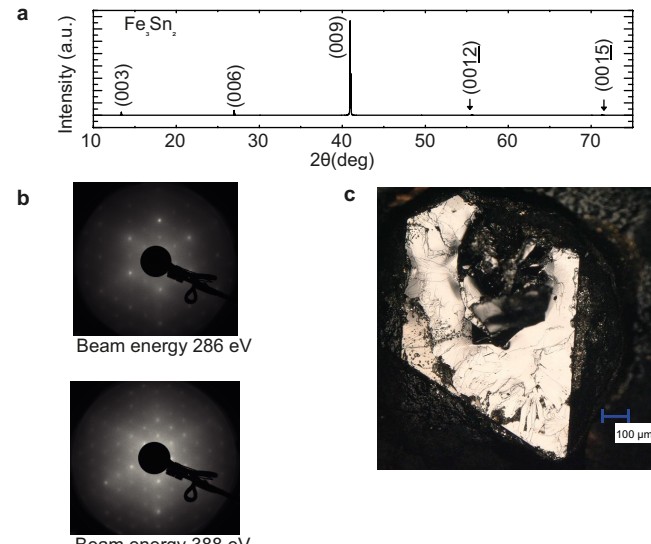

Beam energy 286 eV

Beam energy 388 eV

100 µm

**Extended Data Fig. 4 | Sample characterization. a**, Powder diffraction pattern of polycrystalline $Fe_3Sn_2$ precursor, showing good agreement with the previously reported $R\bar{3}m$ structure, with crystal axes **a**, **b** = 5.34 Å, **c** = 19.80 Å and γ = 120°. **b**, Low-energy electron diffraction of the cleaved surface of an $Fe_3Sn_2$ flake showing the correct symmetry pattern exposing (001) surface or *ab* plane in conventional lattice. **c**, Typical cleaved surface picture of $Fe_3Sn_2$ showing a relatively flat surface suitable for ARPES measurement.

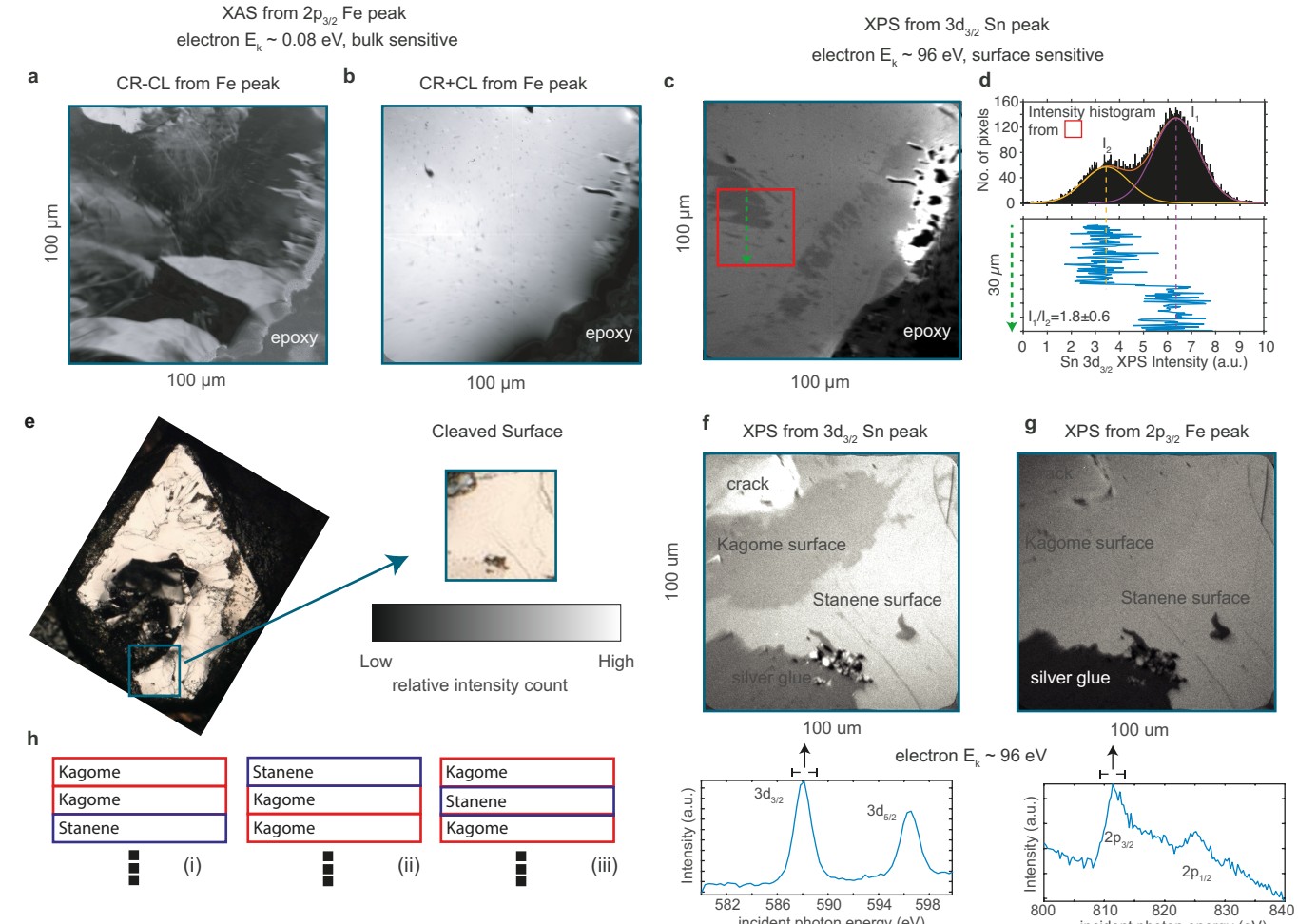

**Extended Data Fig. 5 | XPEEM results. a**, XPEEM magnetic contrast (CR-CL intensity, $T = 80$ K) map taken at the Fe $L_3$ peak, whose signal is 'bulk' sensitive, showing a complex magnetic domain configuration in an area in which panel **b** (CR + CL intensity) shows no contrast variation, indicating a homogeneous phase of the sample. **c**, XPEEM XPS map of the sample at the Sn $3d_{3/2}$ peak, whose signal is surface sensitive, showing a domain structure distinct from the magnetic domain shown in panel **a**, which is attributed to different cleaving terminations, that is, two different surface terminations. This shows that the magnetic domains have no correlation to the surface terminations.

**d**, The intensity difference found in **c** is step-like with intensity ratio of $1.8 \pm 0.6$. **e**, Optical image of a cleaved surface of $Fe_3Sn_2$ single crystal with highlighted area in which the spatially resolved XPS was taken, showing the area with crack and silver glue residue. **f**, Real-space XPS intensity distribution of the Sn $3d_{3/2}$ photoemission line with the typical XPS spectrum shown below it. **g**, Real-space XPS intensity distribution of the Fe $2p_{3/2}$ photoemission line with the typical XPS spectrum shown below it. **h**, The three possible surface terminations.

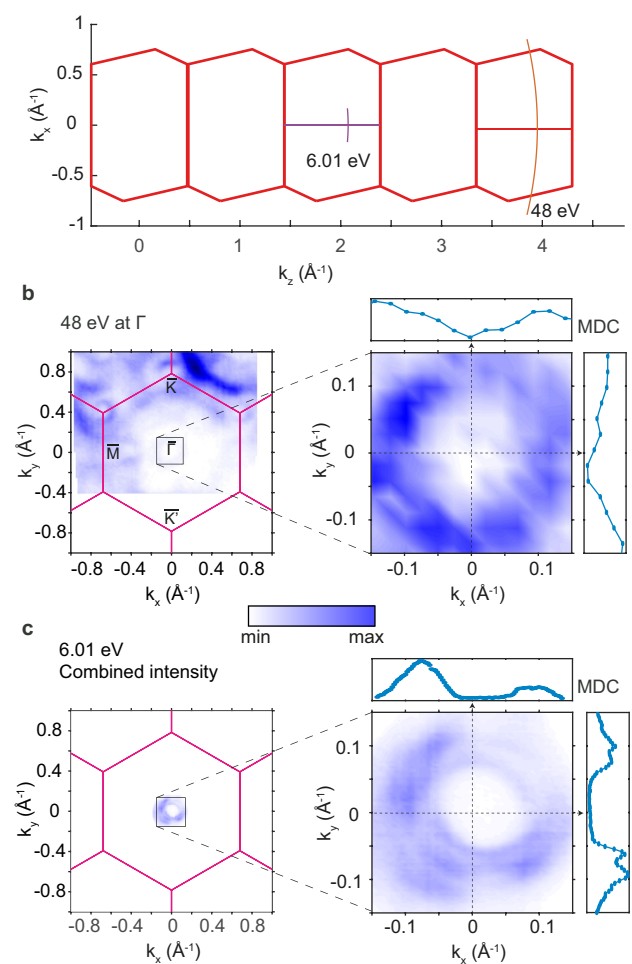

**Extended Data Fig. 6 | Comparison of synchrotron-based and μ-focus laser-based ARPES. a**, Perpendicular momentum ($k_z$) position for 6.01 eV laser energy and 48 eV synchrotron photon energy, showing that the ARPES data were collected on planes with $k_z \approx i \times \frac{2\pi}{c}$. **b**, FS of $Fe_3Sn_2$ at $k_z \approx 0$ for ARPES data using synchrotron light ($hv = 48$ eV). **c**, Combined FS (LH + LV polarization) at $k_z \approx 0$ from twinned-domain (two domains combined) laser ARPES data, 6.01 eV, simulating the synchrotron data probed with a larger beam spot.

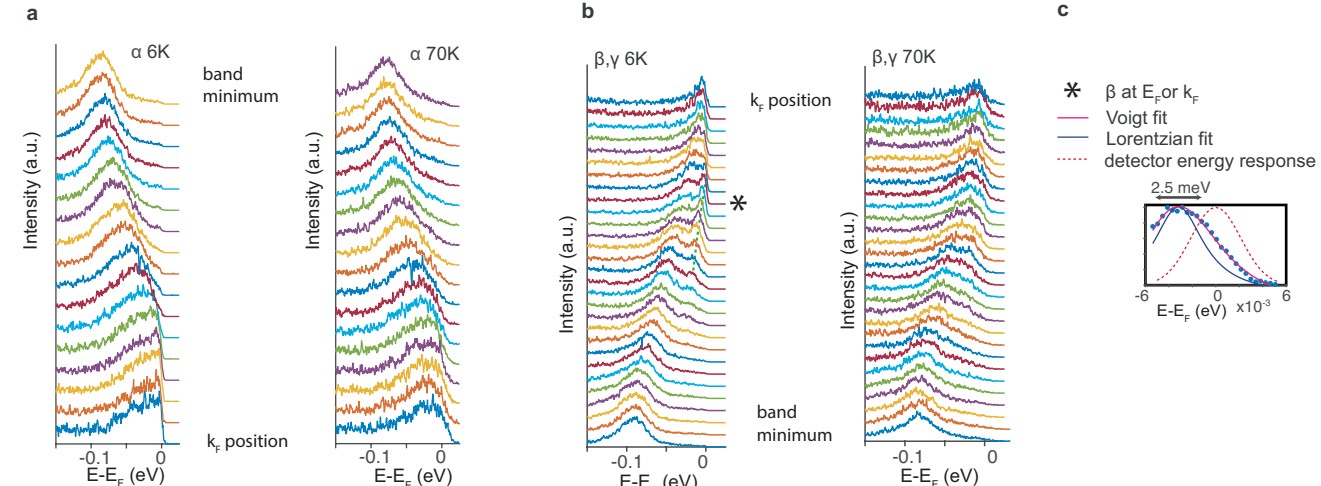

**Extended Data Fig. 7 | Summary of energy distribution curves (EDCs). a,b**, EDC of α (**a**) and β and γ (**b**) at 6 K and 70 K. **c**, Fitting of quasiparticle peak right at $E_F$ (fitting the EDC) by convolving the Lorentzian peak (electron spectral function) and Gaussian peak (detector response).

**Extended Data Table 1 | Summary of Sn relative intensity**

| $I_{\mathrm{Sn},S_s}$ | $I_{\mathrm{Sn},S_{kk}}$ | $I_{\mathrm{Sn},S_k}$ | $\lambda$ (Å) |
|---|---|---|---|
| 2.15 | 1.11 | 1.29 | 1 |
| 2.575 | 1.56 | 1.95 | 2 |

| $I_{\mathrm{Sn},S_s}/I_{\mathrm{Sn},S_{kk}}$ | $I_{\mathrm{Sn},S_s}/I_{\mathrm{Sn},S_k}$ | $I_{\mathrm{Sn},S_k}/I_{\mathrm{Sn},S_{kk}}$ | $\lambda$ (Å) |
|---|---|---|---|
| 1.94 | 1.67 | 1.16 | 1 |
| 1.65 | 1.32 | 1.25 | 2 |

Summary of Sn relative intensity coming from various terminations, $S_s$ (stanene), $S_k$ (single kagome), or $S_{kk}$ (double kagome), with two different electron escape depth values ($\lambda$(Å)).

**Extended Data Table 2 | Summary of the calculated quantities (at 6 K)**

| Band | FS Area (Å⁻²) | dHvA frequency (T) | Fermi velocity $v_F$ (Å·s⁻¹/ℏ) | Effective mass (m*/mₑ) | Scatt. length λ (Å) |
|---|---|---|---|---|---|
| $\alpha$ (Fig. 2) | $0.015 \pm 0.005$ | $157 \pm 20$ | $1.62 \pm 0.05$ | $0.36 \pm 0.06$ | $85 \pm 5$ |
| $\beta$ (Fig. 2) | $0.024 \pm 0.005$ | $254 \pm 20$ | $0.78 \pm 0.01$ | $1.01 \pm 0.08$ | $155 \pm 20$ |
| $\gamma$ (Fig. 2) | $0.041 \pm 0.002$ | $433 \pm 20$ | $0.79 \pm 0.04$ | $1.10 \pm 0.10$ | $38 \pm 5$ |
| $\beta$ (Fig. 4) | N. A. | N. A. | $0.69 \pm 0.05$ | $1.43 \pm 0.03$ | $143 \pm 20$ |
| $\gamma$ (Fig. 4) | N. A. | N. A. | $0.67 \pm 0.05$ | $1.42 \pm 0.03$ | $42 \pm 5$ |

FS area, dHvA frequency, Fermi velocity and scattering length, calculated at the FS of the corresponding α, β and γ bands. N.A. is because of incomplete measured FS map for the high-statistics data.