## [Peer Review File · Nature]

Manuscript Title: Anomalous electrons in a metallic kagome ferromagnet

Reviewer Comments & Author Rebuttals

Redactions – unpublished data

Reviewer Reports on the Initial Version:

Referees' comments:

Referee #1 (Remarks to the Author):

I have read the manuscript by Ekahana et al. with great interest. The experimental manuscript reports laser-based angle-resolved photoemission spectroscopy data on an interesting "correlated" material of great contemporary interest, and claims to find evidence of non (marginal) Fermi liquid-like physics for at least one of the pockets. The core piece of experimental work appears to be interesting and is likely to be of interest to experts in the field. However, I am unconvinced that the manuscript as it stands and with its current research scope is suitable for the wide readership associated with Nature. In my opinion, there is much more that one would like to know about this material, especially with regards to how it fits in with the general scheme of "classifying" other known strongly correlated metals (and non-Fermi liquids, in particular) before I can justify why this article belongs in Nature. Moreover, I have personally found the manuscript to not be very reader friendly (and honestly, a bit confusing), to the point that I feel that this belongs in a specialized journal specifically for quantum condensed matter physicists working once appropriate revisions have been made.

In addition, I have a number of other questions:

- 1) The claims of marginal Fermi-liquid-like behavior seem to arise from the data presented in Fig. 4. I would have thought that the authors would demonstrate this by showing the evidence for "local" criticality associated with the many-body self-energy and relatedly the ω/T scaling. However, I am left confused by how the current data indicates these features.
- 2) I have also been confused by the focus on the electron-pocket. Given that presumably these pockets are not completely decoupled from the other Fermi surfaces in the material, what can we infer about the spectroscopic response associated with the latter? Is it difficult to say whether the claimed marginal character is "infected" to the other pockets as well? This is especially relevant for questions related to electrical transport, of which there isn't much discussion in the manuscript itself. Do these electron pockets contribute to transport at all? I have the same question about thermodynamics (specific heat for instance), which should probe properties of all Fermi surfaces.
- 3) There is discussion of quantum oscillations associated with the same pocket at low temperatures. However, I couldn't find any discussion of the actual temperature dependence of the oscillation amplitude (even though m^* has been extracted). If it were truly marginal Fermi liquid like, I would expect there to be deviations from the standard Lifshitz-Kosevich behavior (there should be additional damping).
- 4) While the apparent agreement between experiment and (DFT+U) is interesting, what is the reader supposed to make of this? Is this a happy coincidence, or is there some important physics issue to be learnt here (that can help direct our future understanding of the "strong" correlation aspect)? Moreover, for the wider readership of a journal like Nature, I would have hoped that the authors would clarify what the related "disagreement" with tight-binding model implies. Naively, tight-binding models are often fit to the DFT bandstructure. Is the point here that just normal DFT is a bad starting point, and one needs to incorporate the DFT "+U" aspect from the outset?

In summary, there are a number of issues associated with the presentation of the current manuscript that makes it unsuitable for Nature, in addition to a more fundamental reason for why the physics associated with a very specific compound is interesting for the wider community. However, if the authors can significantly rewrite their manuscript to improve its clarity and address the above points, I can reconsider my opinion.

Referee #2 (Remarks to the Author):

The manuscript presents the unexpected three-fold symmetric electronic structure of the breathing Kagome lattice Fe_3Sn_2 using a spatially-resolved ARPES experiment. The data measured by XPEEM (XMCD, XPS) and micro ARPES serve numerous information for the researcher curious about the Kagome lattice system. The most important result is that the strong correlation might drive marginal Fermi-liquid behaviour and unexpected in-gap states. The manuscript is written nicely to follow the rigorous approaches and opens a question for the origin of the anomalous magnetotransport properties in Fe_3Sn_2 . This question impacts the study of the correlation and topologies in condensed matter physics, and I am happy to recommend publishing this manuscript. I have a few minor comments below,

- (1) The lines 88-89 in the manuscript, I am curious that previous ARPES data is so far from the signature of the electronic states in Kagome crystals. Because adding the explanation will disturb the narrative of the main text, it would be better to explain in the Supporting information.
- (2) Technical question: How can you make consistency between figure 3a and figure 3c? Because the caption for figure 3c confuses the reader: analysis of spatial map from (i) micro ARPES or (ii) magnetic domain. If figure 3c is the analysis of micro ARPES data, explaining the method in supporting information would be polite to people curious about spatially-resolved ARPES experiments.

Referee #3 (Remarks to the Author):

The manuscript by Ekahana et al., "Anomalous quasiparticle in the zone center electron pocket of the kagomé ferromagnet Fe_3Sn_2 ", exploit laser-based ARPES to get spectroscopic information from a single material's domain. The authors detect three-fold symmetric electron pocket which is in agreement with the DFT calculations and that deviates from the Fermi liquid behaviour. They argue also that these results might follow from electron-electron interaction. While the manuscript is of very high quality and I think that the work should be published, before I can recommend it, there are still a number of major points that I would like the authors to address. I believe that many of these points can be addressed easily and will help improving the readability of this work, thus I hope the authors will appreciate them.

- 1) The Fermi liquid behavior describes the theoretical model of the normal state of electrons in metals in the limit of low temperatures. Away from this limit it is not certain that the electron should follow a Fermi liquid behavior. The data have been collected at 6K. Can the author quantitatively describe, how valid is this assumption on which their entire work is based? From a theoretical perspective, this is not obvious to me.
- 2) The work (cited a few times in the manuscript) by Ye, Kang et al., reporting previous observations of the ARPES of Fe_3Sn_2 detects a gapped Dirac cone at the exact K point (figure 3f of their manuscript) and I don't find it to be too different from the DFT of figure SI10 ($U=0$) that the authors use here. Maybe, I might be missing the point but as it is I don't really find obvious where I should look at in the figures from DFT. Can the authors help the reader for guiding the

visualization of the spectroscopic features? At the moment, I find it only semantic as a difference: Ye, Kang et al. report a gapped Dirac cone, thus not a Dirac cone in the conventional sense with a degeneracy point, which I don't find in disagreement with this work.

3) To prove features at larger k-values, have the authors tried micro-ARPES at the synchrotron? This, correct me if wrong, should allow the authors to use high energy, thus reaching the K point, still maintaining the spot on a single domain. I believe that of the authors want to stress the differences at K, they should have at least an experimental proof.

4) Line 98: "In lower resolution measurements". What does this mean? That their resolutions were smaller? Also, which resolutions?

5) SI14: ARPES vs laser-ARPES. I do agree with the authors that the small light-spot size is a great advantage to remain on a single domain. But I still have difficulties to understand: First of all, the synchrotron data presented in SI14 seems to be much broader than the ones reported in fig. 3a of the manuscript by Ye, Kang et al. Second, from fig SI12, it is clear that the areas belonging to a single termination are $> 100 \text{ um}^2$, which most synchrotron have as a lateral resolution. Can the author comment on this, or be clearer in the main text?

Also from fig.3 of this work, seems like that each twinned domain is of the order of 50 um^2 at least.

6) Why is the finding of rotational domains so important? Isn't this quite common to many materials?

7) The three-fold symmetric pattern: in photoemission, one measures the final state. What happens to the symmetry of this pattern by changing photon energy? I think that would be beneficial to use a second photon energy to see if this pattern is retained, for example 11 eV?

Some minor points:

- I know the wording "triangular kagome" has been used already but I find this imprecise as it should be trihexagonal tilting.

- kagome comes from Japanese and it should not have an accent (i.e. kagomé)

- line 66, "they" it is not clear what the subject is

- I understand what the authors want to say in most parts. However, I believe that to reach the audience of nature, they should make the text much clear. At the moment is extremely complicated to read and follow.

- The order of the SI figures is not easy to follow: the first figure mentioned in the text is SI10. It would be better, for reading reasons, to be chronological.

Author Rebuttals to Initial Comments:

Referee #1 (Remarks to the Author):

I have read the manuscript by Ekahana et al. with great interest. The experimental manuscript reports laser-based angle-resolved photoemission spectroscopy data on an interesting “correlated” material of great contemporary interest, and claims to find evidence of non (marginal) Fermi liquid-like physics for at least one of the pockets. The core piece of experimental work appears to be interesting and is likely to be of interest to experts in the field. However, I am unconvinced that the manuscript as it stands and with its current research scope is suitable for the wide readership associated with Nature. In my opinion, there is much more that one would like to know about this material, especially with regards to how it fits in with the general scheme of “classifying” other known strongly correlated metals (and non-Fermi liquids, in particular) before I can justify why this article belongs in Nature. Moreover, I have personally found the manuscript to not be very reader friendly (and honestly, a bit confusing), to the point that I feel that this belongs in a specialized journal specifically for quantum condensed matter physicists working once appropriate revisions have been made.

Ans: We thank the referee #1 for her/his great interest as well as the recommendation to revise the manuscript. Although referees 2 and 3 consider the manuscript well-written (“written nicely”, “very high quality”), we have made many improvements to the structure and phrasing, paying particular attention throughout the manuscript to fitting the material into the general classification of strongly correlated metals. For example, we now start the paper with explicit statements that there are two routes to strong interaction physics and deviations from the conventional Fermi liquid theory.

In addition, I have a number of other questions:

1) The claims of marginal Fermi-liquid-like behavior seem to arise from the data presented in Fig. 4. I would have thought that the authors would demonstrate this by showing the evidence for “local” criticality associated with the many-body self-energy and relatedly the ω/T scaling. However, I am left confused by how the current data indicates these features.

Ans: We thank the referee for the thoughtful comment. Marginal Fermi liquid behaviour is defined as a direct (linear) proportionality between the imaginary and real parts of the quasiparticle self-energy in the zero temperature limit, and this is indeed consistent with what we observe at small $|k - k_F|$ for the α and β bands after taking into account the finite elastic scattering length measured at the Fermi surface in the form of the constant offsets of the linear Δk versus $|k - k_F|$. Δk for the α and (at very low binding energies) β bands agrees with ω/T scaling: $\Delta k(E, T) = T f\left(\frac{E}{T}\right)$, where $f\left(\frac{E}{T}\right) = (bk_F \left|\frac{E}{\hbar v_F k_F}\right| + c)^\zeta$ and $\zeta = 1$ for marginal Fermi liquid behavior. However, such low energy ω/T analysis misses the important feature of the β band emerging on cooling from the γ band below the very well-defined, finite energy of 20 meV.

2) I have also been confused by the focus on the electron-pocket. Given that presumably these pockets are not completely decoupled from the other Fermi surfaces in the material, what can we infer about the spectroscopic response associated with the latter? Is it difficult to say whether the claimed marginal character is “infected” to the other pockets as well? This is especially relevant for questions related to electrical transport, of which there isn’t much discussion in the manuscript itself. Do these electron pockets contribute to transport at all? I have the same question about thermodynamics (specific heat for instance), which should probe properties of all Fermi surfaces.

Ans: We thank the referee for the electron pocket discussion. The focus on the electron pockets follows for two reasons. First, earlier lower resolution, large laser spot size ARPES measurements on twinned crystals have examined the entire Brillouin zone, yielding very little detail concerning quasiparticles in the bulk material at the Fermi surface. To escape these constraints, we have performed the first high resolution, small spot size ARPES, which has enabled the imaging of sharp quasiparticles at the Fermi surface in *single* twin domains, but at the price of using an optical laser whose long wavelength limits the reachable electron momenta. Second, the pockets are very important and interesting because (a) they are predicted by DFT for Fe₃Sn₂ but not by simple models (such as tight binding models) of kagome bilayers, and (b) they show remarkable evolutions, including the appearance of a new (β) band exactly in the temperature regime of unaccounted-for features in the electrical transport of the material.

Concerning “infection” of other bands: this can occur due to electron-electron (either Coulomb or via phonons) interactions, but to which order in binding energy will depend also on symmetry considerations. Future experiments about the magnitude and visibility of such effects elsewhere in the Brillouin zone are very desirable but will be difficult given the challenging resolution and short electron escape depths associated with going into the VUV regime, and therefore, they are beyond the scope of the present paper

In addition, we thank the referee for bringing up transport and thermodynamics. The electron pockets should contribute to transport and thermal properties and it is true that we do not present any detailed discussion regarding these because they are beyond the scope of this paper. However, we would like to share with the referee our unpublished magnetoresistance (MR) data that shows a linear in field low temperature magnetoresistance (MR) in the figure below. The linear behavior is consistent with marginal Fermi liquid behaviour in Fe₃Sn₂. The temperature dependence of the electrical resistance of Fe₃Sn₂ at $T \leq 40$ K is cubic, suggesting, that the small number of carriers in the zone center pockets are “more” non-Fermi liquid like than the majority elsewhere, but are particularly affected by reachable magnetic fields.

Redactions – unpublished data

3) There is discussion of quantum oscillations associated with the same pocket at low temperatures. However, I couldn't find any discussion of the actual temperature dependence of the oscillation amplitude (even though m^ has been extracted). If it were truly marginal Fermi liquid like, I would expect there to be deviations from the standard Lifshitz-Kosevich behavior (there should be additional damping).*

Ans: We thank the referee for mentioning the quantum oscillations expected from the electron pocket. The referee is correct – there will be deviations from standard Lifshitz-Kosevich behaviour, and we have added a sentence concerning this to the main manuscript as well as more detail to the supplementary information (SI). In particular, the temperature (T) dependence of the oscillations will be determined by (a) the effective mass of the carriers and (b) the effective Dingle temperature, which is conventionally considered a constant, but for a marginal Fermi liquid it would acquire a linear temperature dependence itself. Given our data and the dHvA results of reference ¹, it is possible to estimate the relative impacts of (a) and (b). In particular, our β band is characterized by a Fermi surface area and effective mass similar to the band labelled $\alpha 1$ in reference ¹, for which the dHvA amplitude decays by 1/e by warming to 7 K from the base temperature of ~ 0.4 K. On the other hand, our data show that a temperature increase of 19 K induces a 90% increase in the inverse scattering length Δk , which would translate into only a 8% increase in the effective Dingle temperature, implying that the marginal Fermi liquid behaviour would be difficult to pin down in the dHvA measurement.

4) While the apparent agreement between experiment and (DFT+U) is interesting, what is the reader supposed to make of this? Is this a happy coincidence, or is there some important physics issue to be learnt here (that can help direct our future understanding of the “strong” correlation aspect)? Moreover, for the wider readership of a journal like Nature, I would have hoped that the authors would clarify what the related “disagreement” with tight-binding model implies. Naively, tight-binding models are often fit to the DFT bandstructure. Is the point here that just normal DFT is a bad starting point, and one needs to incorporate the DFT “+U” aspect from the outset?

Ans: We thank the referee for the discussion about the DFT+U in our manuscript and the disagreement with the tight-binding model. To answer the first question, we bring to the referee’s attention that our research *determines* the U value (to be 1.3 eV) by matching the electron pocket minimum obtained from our ARPES data to the bands predicted by DFT+U. This is as “happy a coincidence” as for any establishment of a theoretical parameter from experimental data, and for the general reader of Nature means that there are substantial many-body effects that are neglected in conventional DFT which starts from the non-interacting electron gas. It also means that we have identified a Hubbard (interaction) term which must be added to the tight binding description of Fe₃Sn₂. Such a term is especially important if there are flat bands near the Fermi level, implying, as the referee states, that “one needs to incorporate the DFT “+U” aspect from the outset”.

The disagreement with published (e.g. in Ref²) tight binding models of Fe₃Sn₂ follows not only from the neglect of U, but also from the fact that they do not include all of the couplings (e.g. to Sn atoms in the kagome layers, or to adjacent Sn layers or kagome bilayers in the 3D structure) required to account even for the existence of the electron pockets at the zone center, let alone their properties.

In response to the referee’s comments, we have reformulated the discussion concerning the LDA+U, DFT, and previous work to delineate more clearly the separate points concerning the appropriate tight binding representation and the importance of U.

In summary, there are a number of issues associated with the presentation of the current manuscript that makes it unsuitable for Nature, in addition to a more fundamental reason for why the physics associated with a very specific compound is interesting for the wider community. However, if the authors can significantly rewrite their manuscript to improve its clarity and address the above points, I can reconsider my opinion.

Ans: We thank the referee for openness to reconsideration and for pointing out the “issues associated with the presentation”, which we have addressed by significantly rewriting the manuscript as described above. The physics associated with specific compounds such as Fe_3Sn_2 are interesting to a wider community because only by looking at real materials can basic conceptual issues be tested, such as the validity of thinking about electrons in real metals behaving as free electrons with renormalized masses. Nature has already published two well-cited papers (reference ² and ³) on other aspects of Fe_3Sn_2 , thus revealing that this particular material whose basic building blocks are kagome planes is itself of interest to a wide community. The major step forward in our work is the demonstration that conventional single electron physics breaks down in a direct measurement of the momentum-resolved electron spectral functions.

Referee #2 (Remarks to the Author):

The manuscript presents the unexpected three-fold symmetric electronic structure of the breathing Kagome lattice Fe₃Sn₂ using a spatially-resolved ARPES experiment. The data measured by XPEEM (XMCD, XPS) and micro ARPES serve numerous information for the researcher curious about the Kagome lattice system. The most important result is that the strong correlation might drive marginal Fermi-liquid behaviour and unexpected in-gap states. The manuscript is written nicely to follow the rigorous approaches and opens a question for the origin of the anomalous magnetotransport properties in Fe₃Sn₂. This question impacts the study of the correlation and topologies in condensed matter physics, and I am happy to recommend publishing this manuscript. I have a few minor comments below,

Ans: We thank the referee for the positive comment summarizing succinctly the highlight of our work and the impact. We also thank the referee for the recommendation to publish our manuscript.

(1) The lines 88-89 in the manuscript, I am curious that previous ARPES data is so far from the signature of the electronic states in Kagome crystals. Because adding the explanation will disturb the narrative of the main text, it would be better to explain in the Supporting information.

Ans: We thank the referee for raising the question about previous ARPES data. Here, we paste the sentence at lines 88 – 89. "... between minima and maxima of surface and bulk bands respectively. In addition, the ARPES data collected so far display neither resolved flat bands nor Weyl nodes. However, there is ...". We assume the referee refers to the following sentence: "In addition, the ARPES data collected so far display neither resolved flat bands nor Weyl nodes.". We have added a sub-section in the supporting information explaining the non-observation of flat bands or Weyl nodes as suggested by the referee, which is copied below.

Flat band and Dirac/Weyl crossing in Fe₃Sn₂

The ARPES data from Fe₃Sn₂ presented to date provide no evidence for the flat band predicted from the tight binding calculations (TBC) originally proposed to describe this material. We along with other authors ⁷ attribute this to the breathing nature of the kagome layers as well as the hopping of the electrons across different kagome layers, which is probable, *i.e.*, the electrons are not confined to the kagome layer only, all of which are ignored for the TBC. In addition to taking into account the three dimensionality, the search for flat bands in kagome materials should be extended beyond the one-electron picture and involve correlation, as shown by our results that a U value of 1.3 eV best matches the measured ARPES data to DFT+ U band structure calculations.

Dirac crossings at K points in triangular, honeycomb, and kagome lattices are common in simple tight binding calculations. However, these crossings can be gapped by spin-orbit coupling or small perturbations such as "breathing" distortions of the kagome planes. Thus, our first approach to understanding this kagome material is to use DFT for the 3D material without making assumptions about 2D confinement of the electron movement. We adjust the U value to match the ARPES data.

The difficulty of experimentally detecting Weyl nodes is because they do not occur at high symmetry points in the Brillouin zone and depend on the U value and magnetization vector. If the ARPES beam covers more than one ferromagnetic domain, it will sample the superposition of different band structures and Weyl nodes, making them difficult to detect. Thus, although DFT gives accurate predictions as to where exactly in k-space the Weyl nodes exist, their large number (see ref ⁸) combined with ferromagnetic domain structure and the presence of surface states makes it difficult to find them with the spot sizes and resolution for typical ARPES.

(2) Technical question: How can you make consistency between figure 3a and figure 3c? Because the caption for figure 3c confuses the reader: analysis of spatial map from (i) micro ARPES or (ii) magnetic domain. If figure 3c is the analysis of micro ARPES data, explaining the method in supporting information would be polite to people curious about spatially-resolved ARPES experiments.

Ans: We thank the referee for the question. We decided to remove the sentence in the caption, line 473 – 476 “The magnetization points along the ab-plane at both temperatures but the magnetic domain size is known to be different at these two temperatures, with large domains at temperatures below 64 K and with relatively smaller domains at 80 K”, and the idea of the sentence was inserted in line 240 – 260 of the original manuscript (251 – 253 in new manuscript) to avoid confusion. Regarding the method to produce figure 3c, we refer to the line 233 - 238 in the original manuscript (237 – 239 in new manuscript) where we describe “Figure 3c is the resulting domain image, where for (alpha) band intensity larger on the left side, we label the sample position in red, otherwise we color it green. In the gradation from green to red, white indicates no intensity difference”.

Referee #3 (Remarks to the Author):

The manuscript by Ekahana et al., "Anomalous quasiparticle in the zone center electron pocket of the kagomé ferromagnet Fe₃Sn₂", exploit laser-based ARPES to get spectroscopic information from a single material's domain. The authors detect three-fold symmetric electron pocket which is in agreement with the DFT calculations and that deviates from the Fermi liquid behaviour. They argue also that these results might follow from electron-electron interaction. While the manuscript is of very high quality and I think that the work should be published, before I can recommend it, there are still a number of major points that I would like the authors to address. I believe that many of these points can be addressed easily and will help improving the readability of this work, thus I hope the authors will appreciate them.

Ans: We thank the referee for the positive response and accurate assessment of the key points of the manuscript and the suggestions to improve the readability of the work. We have addressed the issues raised below and hope that the referee finds our revised manuscript suitable for publication in Nature.

1) The Fermi liquid behavior describes the theoretical model of the normal state of electrons in metals in the limit of low temperatures. Away from this limit it is not certain that the electron should follow a Fermi liquid behavior. The data have been collected at 6K. Can the author quantitatively describe, how valid is this assumption on which their entire work is based? From a theoretical perspective, this is not obvious to me.

Ans: We thank the referee for the question of the low temperature limit for describing the Fermi liquid behavior and the relation to our base temperature of 6 K. First, 6 K is equivalent to 0.5 meV, which is considerably smaller than both the energy resolution (~5 meV FWHM) and binding energy range (40 meV) of primary concern in this manuscript. Therefore, our 6 K data lie in the low-temperature limit, although of course raising the temperature moves us into the thermal regime for resolvable low energies.

2) The work (cited a few times in the manuscript) by Ye, Kang et al., reporting previous observations of the ARPES of Fe₃Sn₂ detects a gapped Dirac cone at the exact K point (figure 3f of their manuscript) and I don't find it to be too different from the DFT of figure S110 (U=0) that the authors use here. Maybe, I might be missing the point but as it is I don't really find obvious where I should look at in the figures from DFT. Can the authors help the reader for guiding the visualization of the spectroscopic features? At the moment, I find it only semantic as a difference: Ye, Kang et al. report a gapped Dirac cone, thus not a Dirac cone in the conventional sense with a degeneracy point, which I don't find in disagreement with this work.

Ans: The topic of the present paper is what occurs near the zone centre, and, as mentioned in the introductory paragraphs, the zone centre electron pockets are found in our DFT calculations but not in the tight-binding model of Ye, Kang et al. Notwithstanding this very obvious failure of the tight-binding model, the referee is correct to ask also about the agreement of the tight-binding and DFT calculations, as well as spectroscopic features, elsewhere in the Brillouin zone, most notably near the K point. All of these points have been discussed, with aids for visualization, in previous literature, including, especially refs. ⁴ and ⁵. The main conclusions are that the features studied by Ye et al. are to a large extent defined by *surface states* (not bulk states, and therefore not representative of the intrinsic band structure of the 3D crystal) and that the "gaps" in the DFT band structure arise from the fact that the kagome lattices in Fe₃Sn₂ are of the "breathing" variety, with contributions also arising from the three-dimensional coupling. To prevent confusion, we now state these facts more explicitly in our manuscript.

3) To prove features at larger k -values, have the authors tried micro-ARPES at the synchrotron? This, correct me if wrong, should allow the authors to use high energy, thus reaching the K point, still maintaining the spot on a single domain. I believe that of the authors want to stress the differences at K, they should have at least an experimental proof.

Ans: We thank the referee for the suggestion to investigate the bands near the K point by trying synchrotron based spatially resolved ARPES. In light of previous work ⁴, whichever photon energy we use within the capability of the current available nano-ARPES synchrotron setups, we will always be confined to VUV photon energies with maximum sensitivity to *surface bands*, implying inability to look at the K point in the bulk (not surface states) for a single domain. It is clear that we need both soft X-rays and micrometer beam spot size, which is currently unavailable in any synchrotron, but which we look forward to using after appropriate instruments are constructed over the next decade. We note again that the present paper is primarily about the electron pockets near the zone center, and that we cannot offer insight about the K points which goes beyond references ⁴ and ⁵, which clearly reach different conclusions than Ye et al².

4) Line 98: “In lower resolution measurements”. What does this mean? That their resolutions were smaller? Also, which resolutions?

Ans: We thank the referee for the question. “Lower resolution” means worse resolution, *i.e.*, larger full widths at half maximum (FWHM) of the instrumental response function. Thus “lower resolution measurements” here refer to the worse, relative to that for laser-based ARPES, energy and momentum resolution for the synchrotron-based measurements in reference ⁴. The superior resolution comes from the compression of the photoelectron energy-momentum phase space due to the smaller 6.01 eV laser energy, thus the need of smaller pass energy, which implies a better energy resolution. The outcome is 3 - 5 meV FWHM for our laser ARPES (pass energy 2 and 5) versus typical >10 meV for pass energy larger than 10, which Ye et al at their 92 eV primary photon energy probably used.

5) SI14: ARPES vs laser-ARPES. I do agree with the authors that the small light-spot size is a great advantage to remain on a single domain. But I still have difficulties to understand: First of all, the synchrotron data presented in SI14 seems to be much broader than the ones reported in fig. 3a of the manuscript by Ye, Kang et al. Second, from fig SI12, it is clear that the areas belonging to a single termination are > 100 μm^2 , which most synchrotron have as a lateral resolution. Can the author comment on this, or be clearer in the main text?

Also from fig.3 of this work, seems like that each twinned domain is of the order of 50 μm^2 at least.

Ans: We thank the referee for appreciating that having a small beam spot is indeed a great advantage to stay on a single domain and also for the comments and questions about the benefit of having a micro-focused beam when the termination and twin domains are in the 50-100 μm size. We believe that the referee is referring to figures SI4 and SI2, instead of SI14 and SI12, since we have only 10 figures in SI. Regarding the data presented in SI4 (old number, new number is SI10) vs the data in fig. 3a of the manuscript by Ye, Kang, et. al, we need to emphasize that our data were collected for an incident photon energy of 48 eV, which probes closer to $k_z \approx 0$, while the data shown in fig. 3a. of Ye, Kang, et.al. come from 92 eV, which probes closer to $k_z \approx \frac{3}{4}\pi$ where $k_z \approx \pi$ is the Brillouin zone boundary. We chose the 48 eV data as it lies on the same k_z plane as the laser ARPES data. Since the probed k_z values are different, we cannot directly compare the observed features as they should show different bulk bands since the electronic structure disperses along k_z .

On the other hand, our raw data are no less sharp than the raw data in Ye, Kang, et.al. paper (shown in their SI) as can be seen in figure 2b and c of our preprint ⁴.

The k_z plane that Ye et al chose, which is also not at a high symmetry plane, does not warrant a 6-fold symmetry as they showed in their figure 3a since the bulk bands should be three-fold as dictated by the crystal structure (space group R-3m) and demonstrated by our laser ARPES. This suggests that their 6-fold band comes from either probing 2 twin domains or surface states, and surface states are sharper than bulk states since they are not subject to k_z broadening. The upper part of their Dirac-like cone, which intersects the Fermi surface, is from *surface states*, which clearly also manifests itself in their photon energy-dependent data in their extended Data Figure 4. Their photon energy-dependent data, as those of ref. ⁴ in the VUV regime, are dispersion-less along k_z , completely contrary to the bulk DFT results, and conclusively demonstrating that surface states dominate for the VUV energy range typically probed in synchrotron ARPES measurements. In summary, we suspect that the presented data in the Ye et al. paper look sharper because they were probing surface states and were subject to post-processing, including symmetrization.

Regarding the domain sizes and spatial resolution, figure SI2 (old number, new number is SI7) indicates different possible surface terminations which do not manifest in our spatial scan in figure 3a and 3c; we conclude that figure 3c comes from one kind of termination, which is by Sn. Our concern from figures 3a and 3c are the twin domains that can be simultaneously probed if we use the synchrotron-based ARPES as is shown in figure SI4 (old number, new number is SI10). Our figure 3c indeed clearly shows that the domain size can be smaller than $50 \times 50 \mu m^2$ and this can easily result in probing multiple twin domains since the beam size is also of the order of $50 \times 50 \mu m^2$. Synchrotron data such as in SI4 (old number, new number is SI10) may therefore suffer from twinning, thus yielding a symmetric shape around $\bar{\Gamma}$.

6) Why is the finding of rotational domains so important? Isn't this quite common to many materials?

Ans: We thank the referee for this question. Our finding of rotational domains in the electronic structure is important because it shows directly that the reduced symmetry (3 fold) of Fe_3Sn_2 relative to idealized kagome planes or bilayers (6 fold) has easily visible consequences for electrons at the Fermi surface. This together with the very existence of the zone center pockets demonstrate that to account for the electronic properties of Fe_3Sn_2 we must make very substantial corrections to the simple kagome description. While rotational domains commonly occur for crystal growth, as far as we are aware, ours is the first report of such twinning/rotational domain imaged by spatially resolved ARPES. The observation of twin domains here is important as it clearly suggests that the previous synchrotron-based ARPES may probe both domains together creating an illusion of a 6-fold symmetry out of two 3-fold patterns that are 60 degrees rotated from each other.

7) The three-fold symmetric pattern: in photoemission, one measures the final state. What happens to the symmetry of this pattern by changing photon energy? I think that would be beneficial to use a second photon energy to see if this pattern is retained, for example 11 eV?

Ans: We thank the referee for this question. By changing the photon energy, we are probing different perpendicular momentum or k_z plane. Thus, we expect to see a change in symmetry from 6-fold at the high symmetry plane of $k_z = 0$ and $k_z = \pm\pi$, to a 3-fold pattern in between them. We appreciate the suggestion to change the photon energy, however, we should note that the 6 eV laser that we use is a single-energy laser source that cannot be tuned to 11 eV. Reaching 11 eV requires acquiring a new laser ARPES instrument. We are looking forward to a time when tuning a

continuous laser from 6 eV and above is easily available for an ARPES setup. In the meantime, our current results are interesting and definitive enough to be published in Nature.

Some minor points:

- I know the wording “triangular kagome” has been used already but I find this imprecise as it should be trihexagonal tiling.

Ans: We thank the referee for the suggestion on the term. We have incorporated the suggested term “trihexagonal tiling” in our main manuscript at line 42-43, where previously it said “triangular kagome lattices”, and now has been updated to “trihexagonal tiling lattices (triangular “kagome”)

- kagome comes from Japanese and it should not have an accent (i.e. kagomé)

Ans: We thank the referee for this suggestion. We have updated our writing to “kagome” both in the main manuscript and the SI.

- line 66, “they” it is not clear what the subject is

Ans: We thank the referee for pointing this out. They means “interactions between electrons”. We have changed the word “they” to “the interactions”.

- I understand what the authors want to say in most parts. However, I believe that to reach the audience of nature, they should make the text much clear. At the moment is extremely complicated to read and follow.

Ans: We thank the referee for this comment, which is similar to remarks of referee 1. We have accordingly made extensive improvements to the manuscript, as detailed particularly in our replies to referee 1. As a result, the manuscript should be much easier to follow.

- The order of the SI figures is not easy to follow: the first figure mentioned in the text is SI10. It would be better, for reading reasons, to be chronological.

Ans: We thank the referee for this suggestion. We have updated the order of the SI figures to follow the chronological appearance. The changes are as follows (line here refers to the original unmodified document line):

- Line 84 “SI10” is changed to “SI2”
- Line 92 “SI7-SI9” is changed to “SI3-SI5”
- Line 169 “SI2” is changed to “SI7”
- Line 268 “SI7” is changed to “SI3”
- Line 270 “SI7” is changed to “SI3”
- Line 314 “SI5” is changed to “SI9”

Following these changes, we also have rearranged the order of the sub-chapter and the rest of the figures in the SI.

- The sub-chapter now reads as follow:
 - A. Density Functional Theory Calculations
 - B. Sample growth and structural characterization
 - C. Magnetic domain and surface termination probed by X-ray photoemission electron microscopy (XPEEM)

- D. Micro-focused Laser Angle Resolved Photoemission Spectroscopy (μ -ARPES)
- "SI6" is changed into "SI1"
- "SI1" is changed into "SI6"
- "SI3" is changed into "SI8"
- "SI4" is changed to "SI10"

List of major changes (line based on new file)

- Title changes
- Abstract changes
- Line 82 – 153 Introduction
- Line 191 – 192
- Line 361 – 375
- Line 405 – 406
- Line 405 – 457 (new section and modified conclusion)
- New figure 4
- Newly added figure 5

- 1 Ye, L. D. *et al.* de Haas-van Alphen effect of correlated Dirac states in kagome metal Fe₃Sn₂. *Nature Communications* **10**, doi:10.1038/s41467-019-12822-1 (2019).
- 2 Ye, L. D. *et al.* Massive Dirac fermions in a ferromagnetic kagome metal. *Nature* **555**, 638–+, doi:10.1038/nature25987 (2018).
- 3 Yin, J. X. *et al.* Giant and anisotropic many-body spin-orbit tunability in a strongly correlated kagome magnet. *Nature* **562**, 91–+, doi:10.1038/s41586-018-0502-7 (2018).
- 4 Yao, M. *et al.* Switchable Weyl nodes in topological Kagome ferromagnet Fe₃Sn₂. *arXiv* **1810.01514** (2018).
- 5 Tanaka, H. *et al.* Three-dimensional electronic structure in ferromagnetic Fe₃Sn₂ with breathing kagome bilayers. *Physical Review B* **101**, doi:10.1103/PhysRevB.101.161114 (2020).

-

Reviewer Reports on the First Revision:

Referees' comments:

Referee #1 (Remarks to the Author):

I have gone through all of the previous reports and the response by the authors. I think this is a fine piece of science, but am yet to be convinced of the novelty of the study and the importance of the conclusions that definitely warrant acceptance in Nature (in light of many uncertainties associated with the interpretation and technical difficulties associated with answering some of the key questions).

Referee #2 (Remarks to the Author):

The authors corrected a manuscript with replies about non-fermi liquids, DFT+U, and spatially-resolved ARPES used in this research. Even though the presented data is not focused on the corners in the surface Brillouin zone, it is enough to make questions about realistic electronic structures in kagome crystals, including flat bands.

The current manuscript will influence researchers interested in electronic structures and related properties of kagome crystals with breathing. Therefore, I am delighted to recommend this manuscript to be published in Nature.

Referee #3 (Remarks to the Author):

I have read the manuscript and the answers very carefully. I think that the authors have done a real effort in answering the referees' comments and many of my doubts find now an explanation. The manuscript itself, however, is still very hard to read and follow. I think the work should be simplified as, in the current form, even if the answers given were convincing, I find it not accessible for the audience of this journal. I hope the authors understand my concerns. I still think this work is suitable for this journal and the science is sound, but I also think that still effort must be made to guarantee a wide readability.

I report some examples of this, for instance at the abstract level,

1) "interactions between the electrons induce a mild degradation - quadratic in deviation from the Fermi surface - of the definition of electron energies but lead neither to a change in the Fermi volumes nor additional spectral peaks at the Fermi surface." I believe I understand what the authors mean by reading the article but I do struggle in understanding what "mild degradation of the definition of..." means. I strongly recommend changing these sentences to make them clearer and simpler.

2) Talking about the 3rd band: "low temperatures and energies". Which energies are the authors talking? Thermal energy? if so, they are stating the same thing twice. Or Binding energy? I find that many points in the manuscript have this sort of statements which are difficult to follow.

3) All of the color scales are missing. This is important for the reader to understand saturation levels, to help the understanding of the analysis, and to make the manuscript more precise. These

must be added in all the images.

4) What is " virtual occupancy"? Is it the possibility of occupying the band via a virtual scattering process? The authors need to be more precise about which are the possible scattering channels different from thermal, i.e., which are the ones allowed and which are not?

5) Fig 4a: fitting at high temperature. The authors show two bands together gamma+beta. How does the Chi squared compare if they use an extra component to account in fact for both bands? I think it would be more realistic to include it and would add precision to the manuscript

I hope the authors will understand. If they make the changes above I do not have any objection against recommending publication.

Author Rebuttals to First Revision:

Referees' comments:

Referee #1 (Remarks to the Author):

I have gone through all of the previous reports and the response by the authors. I think this is a fine piece of science, but am yet to be convinced of the novelty of the study and the importance of the conclusions that definitely warrant acceptance in Nature (in light of many uncertainties associated with the interpretation and technical difficulties associated with answering some of the key questions).

We thank the referee for the comment that our work is a fine piece of science. We emphasize here that the novelty of the study is the discovery of a state (β) which is made of borrowed quasiparticles from a nearby (energy and momentum-wise) band (γ) due to its interaction with the flatband. This conclusion is important in the study of kagome systems where flatbands have been predicted based on simple tight binding models and expected to play a significant role in their electronic properties. Our study here is the first to visualize the impact of flatband physics in the formation of another band.

Referee #2 (Remarks to the Author):

The authors corrected a manuscript with replies about non-fermi liquids, DFT+U, and spatially-resolved ARPES used in this research. Even though the presented data is not focused on the corners in the surface Brillouin zone, it is enough to make questions about realistic electronic structures in kagome crystals, including flat bands.

The current manuscript will influence researchers interested in electronic structures and related properties of kagome crystals with breathing. Therefore, I am delighted to recommend this manuscript to be published in Nature.

We thank the referee for the recommendation for our work. The corner of the Brillouin zone will be discussed separately in a separate manuscript.

Referee #3 (Remarks to the Author):

I have read the manuscript and the answers very carefully. I think that the authors have done a real effort in answering the referees' comments and many of my doubts find now an explanation. The manuscript itself, however, is still very hard to read and follow. I think the work should be simplified as, in the current form, even if the answers given were convincing, I find it not accessible for the audience of this journal. I hope the authors understand my concerns. I still think this work is suitable for this journal and the science is sound, but I also think that still effort must be made to guarantee a wide readability.

We thank the referee for the recommendation of our work. We appreciate the suggestion to modify the manuscript to suit the audience of Nature and to make it more readable. We have updated our manuscript accordingly.

I report some examples of this, for instance at the abstract level,

1) "interactions between the electrons induce a mild degradation - quadratic in deviation from the Fermi surface - of the definition of electron energies but lead neither to a change in the Fermi volumes nor additional spectral peaks at the Fermi surface." I believe I understand what the authors mean by reading the article but I do struggle in understanding what "mild degradation of the definition of..." means. I strongly recommend changing these sentences to make them clearer and simpler.

We thank the referee for the suggestion. By “mild degradation of the definition of electron energies”, we meant the inverse of the quasiparticle lifetime being quadratic in the deviation of E from E_F . We have changed that into

“Quasiparticle decay rate is quadratic in the deviation of the binding energy from the fermi level.”

2) Talking about the 3rd band: "low temperatures and energies". Which energies are the authors talking? Thermal energy? if so, they are stating the same thing twice. Or Binding energy? I find that many points in the manuscript have this sort of statements which are difficult to follow.

We thank the referee for the comment. The energies refer to the binding energies. We have stated explicitly “binding energies”.

3) All of the color scales are missing. This is important for the reader to understand saturation levels, to help the understanding of the analysis, and to make the manuscript more precise. These must be added in all the images.

We thank the referee. We have put the color-bars as suggested.

4) What is " virtual occupancy"? Is it the possibility of occupying the band via a virtual scattering process? The authors need to be more precise about which are the possible scattering channels different from thermal, i.e., which are the ones allowed and which are not?

We thank the referee for the question. Yes, virtual occupancy here means the possibility of occupying the band via a virtual scattering process. However, we have greatly modified the manuscript and have removed those words.

5) Fig 4a: fitting at high temperature. The authors show two bands together gamma+beta. How does the Chi squared compare if they use an extra component to account in fact for both bands? I think it would be more realistic to include it and would add precision to the manuscript

We thank the referee for the comment. Fitting with two peaks will be dominated by one component and drive the other component to be insignificant, making it effectively a single peak fitting. Below we show our results for

$$\chi^2 = \sum_i \frac{(x_i - x_{fit,i})^2}{\sigma_i^2}$$

where x_i is the intensity data obtained from the experiment, $x_{fit,i}$ is the fitted value at position i (momentum i), and σ_i is the standard deviation of the obtained intensity. Since we did not do a time series measurement to calculate σ_i , we instead consider the noise of the data where the intensity should be zero as σ_i and we assume σ_i to be a constant for every energy position i . In practice, we can compare the following value between the two fittings

$$\chi^2 \propto \sum_i |x_i - x_{fit,i}|^2$$

The result of the comparison between a single peak vs two peaks at each energy level is given below

We can see that the result for 2 peaks fitting consistently has higher variances than the 1 peak fitting. Thus, the 1 peak model is more acceptable than forcing the signal to be 2 peaks.

I hope the authors will understand. If they make the changes above I do not have any objection against recommending publication.

We thank the reviewer for the detailed suggestions.